# RAU: REFERENCE-BASED ANATOMICAL UNDERSTANDING WITH VISION LANGUAGE MODELS

## ABSTRACT

Anatomical understanding through deep learning is critical for automatic report generation, intra-operative navigation, and organ localization in medical imaging; however, its progress is constrained by the scarcity of expert-labeled data. A promising remedy is to leverage an annotated reference image to guide the interpretation of an unlabeled target. Although recent vision–language models (VLMs) exhibit non-trivial visual reasoning, their reference-based understanding and fine-grained localization remain limited. We introduce RAU, a framework for reference-based anatomical understanding with VLMs. We first show that a VLM learns to identify anatomical regions through relative spatial reasoning between reference and target images, trained on a moderately sized dataset. We validate this capability through visual question answering (VQA) and bounding box prediction. Next, we demonstrate that the VLM-derived spatial cues can be seamlessly integrated with the fine-grained segmentation capability of SAM2, enabling localization and pixel-level segmentation of small anatomical regions, such as vessel segments. Across two in-distribution and two out-of-distribution datasets, RAU consistently outperforms a SAM2 fine-tuning baseline using the same memory setup, yielding more accurate segmentations and more reliable localization. More importantly, its strong generalization ability makes it scalable to unseen datasets, a property crucial for medical image applications. To the best of our knowledge, RAU is the first to explore the capability of VLMs for reference-based identification, localization, and segmentation of anatomical structures in medical images. Its promising performance highlights the potential of VLM-driven approaches for anatomical understanding in automated clinical workflows.

## 1 INTRODUCTION

Anatomical understanding is critical in medical image analysis (Schmidt et al., 2024), serving as a foundational component (Li et al., 2025) for a wide range of critical applications, such as automated report generation (Wang et al., 2025), intraoperative navigation (Khan et al., 2024), and organ localization (Xu et al., 2024), and thus supporting accurate diagnostics (Hartsock & Rasool, 2024), effective treatment planning (Gao et al., 2025), and precise surgical execution (Gaudioso et al., 2024). Traditionally, each of these tasks often requires the design and training of dedicated models tailored to their specific objectives (van Veldhuizen et al., 2025; Alozai et al., 2025). However, the scarcity of high-quality, annotated medical imaging datasets poses a significant challenge (Wang et al., 2024a), since the acquisition of such annotations is resource-intensive and requires domain expertise (Jin et al., 2023). This shortage of labeled data substantially hinders the ability to train robust and generalizable models for anatomical understanding (Bian et al., 2025), underscoring the need for approaches that can mitigate dependence on large-scale manual annotation (Fan et al., 2025).

Multimodal large language models (MLLMs) offer a promising solution to this challenge due to their strong learning and generalization abilities (Hu et al., 2024; Nam et al., 2025). Numerous studies have demonstrated that vision–language models (VLMs) can enhance their understanding of visual content through targeted training strategies (Li et al., 2023; Tanno et al., 2025). For instance, supervised fine-tuning (SFT) can be employed to align the model's outputs with high-quality, task-specific annotations, thereby improving accuracy in domain-relevant recognition tasks (Tanno et al., 2025; Li et al., 2023; Chen et al., 2024c). Reinforcement learning (RL), particularly methods such as Group Relative Policy Optimization (GRPO) (Shao et al., 2024), empowers VLMs with

stronger reasoning capabilities by encouraging explicit chain-of-thought (CoT) generation, which not only enables complex multimodal inference but also drives superior generalization across both in-domain and out-of-domain data (Shao et al., 2024; Guo et al., 2025). Furthermore, few-shot learning paradigms exploit a small number of annotated exemplars, in-context learning (ICL) enables VLMs to adapt to new anatomical queries directly from reference demonstrations at inference time, and retrieval-augmented generation (RAG) enriches the model's contextual grounding by incorporating relevant external knowledge (Lian et al., 2024; Dutt et al., 2024; Liu et al., 2025a; Yang et al., 2025; Xia et al., 2025). These techniques are particularly beneficial for medical image understanding, where human anatomy is largely shared across individuals, as they collectively mitigate the impact of scarce high-quality annotations while preserving adaptability and robustness across diverse clinical scenarios.

Despite recent progress, medical VLMs have so far approached anatomical understanding primarily as a knowledge- and pattern-matching problem (Chen et al., 2024b; Xu et al., 2025; Pan et al., 2025): with sufficient supervised data, they can generalize across tasks and output formats (VQA, bounding boxes, or even segmentation when coupled with vision foundation models (Lai et al., 2024; Liu et al., 2025b)), but they remain heavily data-hungry and their reasoning is largely constrained by the medical priors accumulated during training (Pan et al., 2025; Huang et al., 2025). In more complex anatomical understanding tasks—such as identifying the correct vessel branch or precisely localizing a stenosis—labels are scarce and target structures often lack distinctive semantic cues (Ma et al., 2022; Choi et al., 2015), so relying solely on encoded medical knowledge and textual reasoning is insufficient. In real clinical workflows, human annotators typically solve such cases by comparing a target scan against a reference image and reasoning about their spatial relationships (Popov et al., 2024). Motivated by this practice, we propose to explicitly steer VLMs toward reference-based spatial relationship reasoning: instead of asking the model to "know" every structure a priori, we provide a small set of high-quality annotated reference images and let the model infer anatomy in the target image by reasoning over their spatial correspondence, thereby enabling more precise and data-efficient anatomical understanding.

Here are the main contributions of this work:

- Inducing reference-based spatial reasoning in VLMs for medical images. We empirically show that targeted training enables a VLM to reason about relative spatial relations with respect to a reference image, reducing the need for large annotated datasets in anatomical understanding.
- Pixel-level anatomical region identification via VLM × SAM2. We show that the VLM's spatial reasoning capability is preserved after integration and co-training with SAM2. Leveraging SAM2's precise segmentation capability, the combined system yields accurate identification of target structures in novel images given a reference image.
- Generalization to Out-of-Distribution (OOD) data with broad applicability. Our method yields consistent improvements on OOD datasets, underscoring its robustness to distribution shifts and suggesting strong potential for downstream use cases including automatic reporting, surgical navigation, and image-guided intervention planning.

## 2 RELATED WORK

### 2.1 ANATOMICAL UNDERSTANDING WITH NON-VLM PIPELINES

Classical anatomical understanding in medical imaging has been dominated by non-VLM pipelines based on registration and fully supervised segmentation. Atlas-based registration warps a patient scan into a standardized anatomical space so that labels can be propagated from an annotated atlas to the target image (Ramadan et al., 2024; Wang et al., 2012; Varol Arısoy et al., 2025). In practice, modern pipelines often combine strong universal segmentation baselines such as nnU-Net (Isensee et al., 2021) or multi-organ foundation models (Berrezueta et al., 2025; Yan et al., 2025; Wasserthal et al., 2023) with learning-based deformable registration modules like VoxelMorph (Balakrishnan et al., 2019). These systems can yield high-quality organ masks in relatively standardized CT or MR protocols and are now widely used as building blocks for downstream tasks.

However, these non-VLM approaches have important limitations when viewed through the lens of concrete clinical and workflow scenarios. In surgical navigation (e.g., determining at which vessel

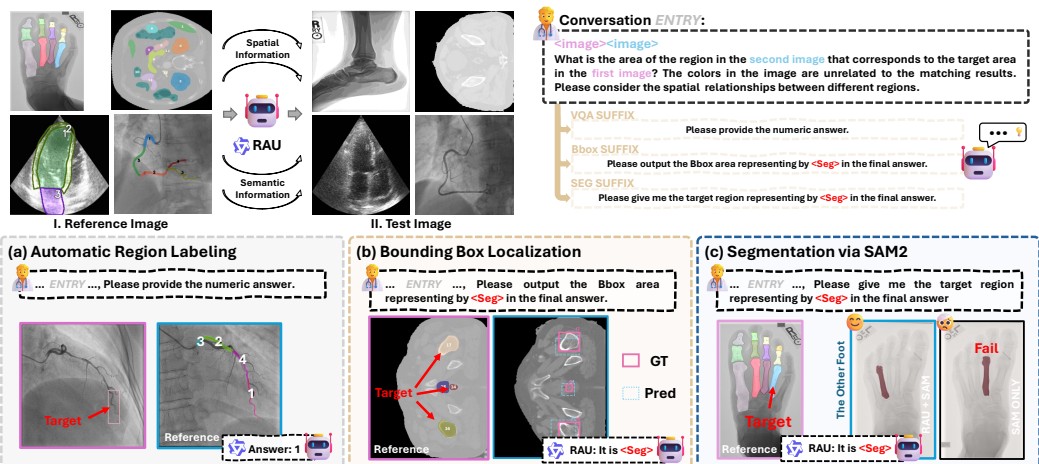

Figure 1: **RAU** enables anatomical understanding with minimal reference. We explore three task modes: (a) VQA, (b) Bbox Localization (c) Segmentation with SAM2 integration. Representative cases are shown in the 2nd row. Additional results and examples are provided in later sections.

segment the wire or catheter tip is located under fluoroscopy), registration to a template mask with ROI labels provides global context but tends to perform unstably due to thin and tubular structures of interventional devices and vessels, as well as motion, artifacts, incomplete views, and rapidly changing surgical scenes (Matl et al., 2017; Zhu et al., 2020). In longitudinal studies, classical registration is a natural tool for tracking the same lesion across timepoints, but performance can degrade under large anatomical changes, inconsistent fields-of-view, or tumor shrinkage/growth, which often require expert intervention (Tanno et al., 2023). As for machine-assisted annotation, automatic masks from nnU-Net (Isensee et al., 2021)/TotalSegmentator or MedSAM (Ma et al., 2024; Wasserthal et al., 2023) can accelerate labeling, yet their data-hungry nature (Wu et al., 2025) and sensitivity to scanner/protocol shifts (Rayed et al., 2024; Liu et al., 2025c) limit deployment across institutions.

Given these limitations, our aim is not to design yet another fully supervised segmentor, but a reference-based framework for anatomical understanding. By guiding a VLM to reason about spatial relationships between a labeled reference image and an unlabeled target, our method (i) achieves strong OOD generalization across modalities and datasets, (ii) remains robust in complex scenes where contrast and texture cues are weak but relative locations are preserved, and (iii) naturally supports multiple output formats (VQA answers, bounding boxes, and segmentation masks) to serve different clinical tasks with minimal extra supervision.

## 2.2 VLM-BASED ANATOMICAL GROUNDING AND ITS LIMITATIONS

VLMs offer a complementary paradigm that demonstrates strong capabilities in free-form image captioning, spatial relational reasoning, and compositional part–whole understanding (Bai et al., 2023; Shen et al., 2022; Chen et al., 2024a). In medical domains, such models have been adapted for image captioning and reporting, as well as for multiple-choice visual question answering (VQA) that aligns with structured exams or board-style questions (Nam et al., 2025; Stogiannidis et al., 2025). Compared to conventional pipelines, VLMs exhibit superior task-level generalization: by leveraging encoded medical priors, a single model can support diverse downstream tasks and generalize more robustly across imaging modalities and acquisition qualities.

Beyond global captioning and VQA, VLMs have been extended to explicit grounding and localization. Open-vocabulary detectors such as GLIP, OWL-ViT, and Grounding DINO treat text as detection queries and output bounding boxes (Li et al., 2022; Alhadidi et al., 2024; Zhang et al., 2022b; Liu et al., 2024c), enabling prompts like "wire tip" or "stenosis" to be mapped to candidate regions. Representative systems such as LISA (Lai et al., 2024) decode special segmentation tokens through SAM/Mask2Former-style backbones (Zhang et al., 2022a), while SEEM and EVF-SAM leverage language-as-queries and promptable segmentation to obtain masks guided by text or

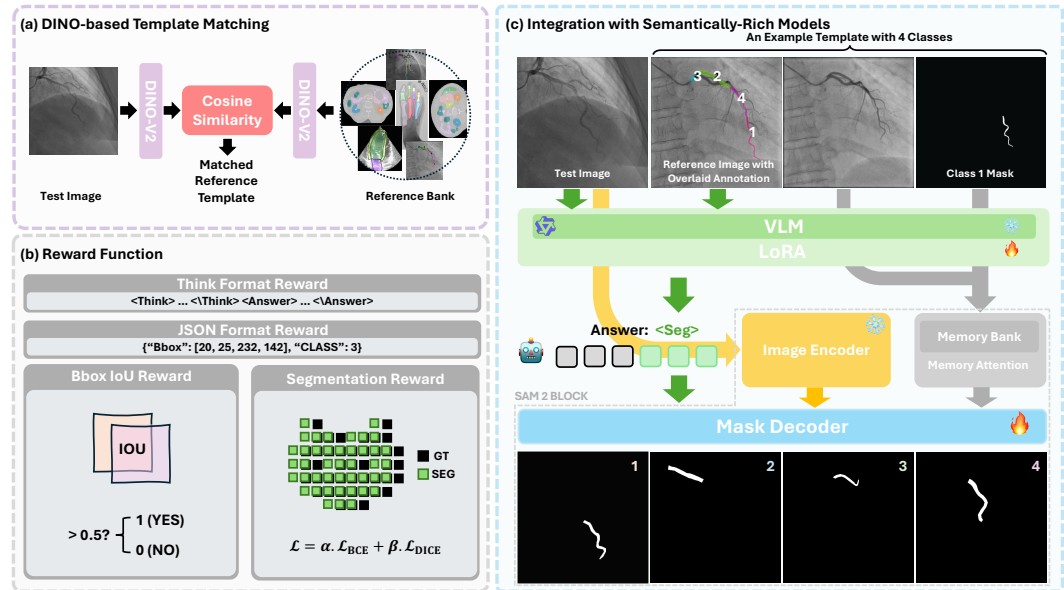

Figure 2: **Method overview**. (a) For a target test image, we extract DINOv2 features and retrieve the best-matching annotated template from a reference bank using cosine similarity. (b) Reinforcement-learned Qwen VLM reads the paired target–reference and generates a special <Seg>token, which is projected via an MLP adapter and injected into the SAM2 decoder as a soft spatial prompt. Reward functions include format validity and segmentation quality (Dice+BCE). (c) SAM2 leverages both the projected VLM embedding and memory attention from the reference mask to guide segmentation on the unlabeled target without explicit box/point prompts.

points (Zou et al., 2023; Zhang et al., 2024). With ICL (Dong et al., 2022) and additional SFT and reinforcement learning (Dong et al., 2023; Liu et al., 2024a), these VLM-based pipelines can, in principle, support a unified interface for surgical navigation, diagnosis, longitudinal tracking, and annotation.

Yet despite recent progress, current VLM-based methods still fall short of clinical-grade anatomical understanding (Li et al., 2024). Regardless of whether the output format is free-form text, bounding boxes, or segmentation masks, the underlying reasoning primarily relies on priors accumulated during pretraining and subsequent SFT/RL, and thus remains confined to the medical knowledge in training data (Bian et al., 2025; Nam et al., 2025). Even when SFT and RL are used to strengthen "logical" chains of thought, this logic typically encodes what should happen in canonical disease presentations, and may still fall short when precise, spatially grounded localization is required. As a result, the best medical VLMs still struggle to achieve reliable anatomical understanding in safety-critical settings: their behavior remains data-hungry and brittle, especially for rare configurations, device interactions, or subtle negative findings (Liu et al., 2024b; Jin et al., 2023). This limitation is partly due to the prohibitive cost of collecting large-scale, pixel-accurate annotations that cover the combinatorial complexity of real clinical workflows, and partly due to the intrinsic diversity and open-endedness of medical imaging environments. Consequently, methods that can impose anatomical structure and remain reliable in specialized applications, even under limited supervision, are especially valuable. Motivated by this, we study a reference-based spatial reasoning paradigm in which a VLM compares an unlabeled target image to a sparsely annotated template to achieve data-efficient anatomical understanding across modalities and tasks.

## 3 ENHANCING ANATOMICAL UNDERSTANDING IN VLMS

In this section, we systematically explore how to best endow VLMs with spatial anatomical understanding. Section 3.1 formulates reference-based anatomical understanding as a VQA task, where the VLM, conditioned on a labeled reference image and an unlabeled target, answers anatomy-aware questions to localize structures. Section 3.2 then upgrades the output space from text to explicit bounding boxes, providing a more flexible and reusable representation for downstream pipelines

than pure VQA-style outputs. However, bounding boxes are inherently suboptimal for thin, curved, or spatially fragmented targets (e.g., vessels or elongated devices). Section 3.3 therefore couples the VLM with a segmentation foundation model and directly predicts dense masks, enabling more precise and fine-grained anatomical understanding.

Across Sections 3.1–3.3, we employ a shared experimental protocol and dataset suite. We train on two labeled datasets (RAOS-CT (Luo et al., 2024), Arcade-X-Ray (Popov et al., 2024)) and evaluate out-of-distribution generalization on multiple external datasets (LERA-X-Ray (Varma et al., 2019), CAMUS-Ultrasound (Leclerc et al., 2019) AMOS2022-MRI, CT-Liver, CT-Lung) that span diverse modalities and anatomical regions; see Section A.2 for details.

## 3.1 Visual Question Answering

We formulate the task in a VQA-style framework by annotating anatomical regions in the reference image and using prompt engineering to guide the VLM in identifying corresponding regions in the target image. An example prompt can be found in Figure 2b. This guides the VLM to perform spatial reasoning by grounding the query image in the labeled reference, thereby facilitating accurate target region identification.

**Methods** Formally, given a reference image $I^{\mathrm{ref}}$ with annotated regions $\{r_1, r_2, \ldots, r_m\}$ and their labels $\{y_1, y_2, \ldots, y_m\}$, and a target image $I^{\mathrm{tgt}}$, the task is to predict the label $\hat{y}$ corresponding to a queried region $r^{\mathrm{tgt}}$ in $I^{\mathrm{tgt}}$. We formulate this VQA interaction (Antol et al., 2015) as:

$$\hat{y} = \arg\max_{y \in \mathcal{Y}} P_\theta\big(y \mid I^{\mathrm{ref}}, \{(r_i, y_i)\}_{i=1}^{m}, I^{\mathrm{tgt}}, r^{\mathrm{tgt}}, \mathrm{prompt}\big), \tag{1}$$

where $P_\theta$ is the VLM parameterized by $\theta$ and $\mathcal{Y}$ is the set of candidate anatomical labels.

We employ a two-stage strategy: first SFT, followed by RL. During SFT, the VLM is trained on labeled reference–target pairs to learn the task formulation and adapt to the medical domain. Later, GRPO is applied to further enhance performance and generalizability . The reward function is defined as a weighted combination of *accuracy* and *format correctness*:

$$R = \lambda_{\mathrm{acc}} \cdot \mathbb{1}\{\hat{y} = y\} + \lambda_{\mathrm{fmt}} \cdot \mathbb{1}\{\text{output format is valid}\}, \tag{2}$$

where $\mathbb{1}\{\cdot\}$ is the indicator function, and $\lambda_{\mathrm{acc}}, \lambda_{\mathrm{fmt}}$ are weighting coefficients. This reinforcement signal explicitly enforces the VLM to produce correct labels in the expected format, thereby fostering robust spatial reasoning ability.

Table 1: Labeling Accuracy via VQA on in-/out-of-domain datasets. Models are trained on RAOS (650k). The number of label classes for each dataset is shown in parentheses next to its name.

| VLM | In-Distribution (ID) | Out-of-Distribution (OOD) | | | | |
|---|---|---|---|---|---|---|
| | RAOS-test-CT (20) | Arcade (26) | CT-Liver (3) | CT-Lung (3) | AMOS2022-MRI (17) | Mixture |
| Qwen2.5VL-7B(Vanilla) | 16.62% | 13.40% | 53.17% | 54.69% | 19.45% | 20.46% |
| MedFlamingo (Moor et al., 2023) | 21.03% | 14.41% | 50.20% | 57.95% | 34.22% | 22.09% |
| Med-VLM-R1 (Pan et al., 2025) | 18.27% | 12.11% | 58.91% | 61.13% | 20.06% | 21.23% |
| HuatuoGPT-Vision (Chen et al., 2024b) | 23.73% | 13.85% | 63.23% | 60.78% | 22.81% | 24.33% |
| Lingshu (Xu et al., 2025) | 22.87% | 14.20% | 61.81% | 59.32% | 19.42% | 23.95% |
| MedRegA (Wang et al., 2024b) | 15.75% | 12.63% | 46.72% | 41.59% | 13.04% | 16.80 % |
| BiRD (Huang et al., 2024) | 16.49% | 15.92% | 57.19% | 51.93% | 16.76% | 22.94% |
| MedPLIB (Huang et al., 2025) | 17.28% | 14.26% | 54.33% | 56.47% | 27.81% | 23.61% |
| SFT (3 Epochs) | 41.62% | 26.88% | 58.51% | 57.70% | 45.52% | 42.16% |
| SFT (5 Epochs) | 64.11% | 33.92% | 77.09% | 75.73% | 67.58% | 64.91% |
| GRPO (800 Steps) | 42.25% | 38.38% | 62.03% | 61.81% | 41.37% | 44.65% |
| GRPO (1600 Steps) | 62.36% | 41.03% | 75.62% | 73.33% | 63.75% | 62.47% |
| GRPO (2400 Steps) | 70.68% | 48.37% | 83.76% | 80.10% | 72.93% | 70.84% |

**Experiment Settings** We first train our reference-based VQA models on RAOS-CT, aiming to teach the VLM to solve the task via spatial anatomical reasoning rather than surface pattern matching. Each training sample consists of a labeled reference image, an unlabeled target image, and a prompt that specifies the anatomical query on the reference. The model must infer the correspondence and produce an answer in a constrained format that includes only its reasoning trace and the final index of the matching region in the reference image (see Appendix A.3.2). After training on RAOS-CT, we evaluate the same model on the RAOS-CT test split and on a suite of out-of-distribution datasets—Arcade-X-Ray, LERA-X-Ray, CAMUS-Ultrasound, AMOS2022-MRI, CT-Liver, and CT-Lung—under the same VQA-style protocol, to assess generalization and robustness across modalities, anatomical regions, and imaging conditions.

**Experiment Results** As shown in Tab. 1, scaling up training with either SFT or RL leads to substantial improvements in labeling accuracy over the vanilla baseline. On the ID dataset RAOS-test-CT, accuracy increases from 16.62% to 64.11% with 5 epochs of training, and further to 70.68% with 2400 steps RL-based finetuning, representing a 54.06% absolute gain.

Training also yields strong cross-dataset generalization. Averaged over the five OOD datasets, accuracy improves from 32.23% with the vanilla baseline to 63.85% with SFT, and further to 71.20% with RL. The RL finetuned model achieves the best score on every OOD dataset: 48.37% on Arcade, 83.76% on CT-Liver, 80.10% on CT-Lung, 72.93% on AMOS, and 70.84% on the Mixture set, consistently surpassing SFT by 4.4 to 14.5 percentage points, depending on the dataset. Within the RL regime, accuracy improves monotonically with additional optimization steps across both ID and OOD settings, suggesting that continued policy optimization enhances generalizable decision-making rather than overfitting to the training distribution.

Qualitative inspection of CoT outputs during training reveals a convergence toward reference-guided relational reasoning, wherein the model increasingly grounds its decisions in spatial relationships between anatomical regions rather than in simple visual features. Examples can be seen in Figure A3. Notably, the poor performance of Med-VLM-R1 (Pan et al., 2025) further corroborates this observation: when reinforcement learning is applied directly for reasoning-based target recognition in medical images, the model often attempts to localize organs without leveraging relational priors, leading to relatively low accuracy. Similarly, MedFlamingo (Moor et al., 2023), HuatuoGPT-Vision (Chen et al., 2024b), and Lingshu (Xu et al., 2025)—large-scale foundation models with stronger medical priors—also perform poorly on this task, indicating that even medically specialized models struggle to achieve robust anatomical understanding in complex settings. This further suggests that reasoning from explicit spatial relationships, rather than directly recognizing organs, is a more principled and effective strategy. Together, we show that VLMs can develop non-trivial spatial understanding and robust generalization in reference-based localization tasks, and that reasoning over spatial relations between different regions of the reference image provides a more generalizable strategy.

However, the presence of many densely arranged targets, such as intestinal loops in abdominal CT (Luo et al., 2024), in the reference image can degrade the accuracy of VQA-based approaches, underscoring the need for more precise and anatomy-aware method.

## 3.2 Bbox Prediction and Global Matching

In this section, we extend the task to bounding box (bbox) prediction. Rather than classifying a region label, the VLM is prompted to directly output the location of the corresponding region in the target image. However, VLMs often hallucinate numerical coordinates (Liu et al., 2023), especially under limited supervision.

**Methods** To address this, we use a two-stage mechanism: 1) the VLM outputs (Lai et al., 2024) one or more special `<Seg>` tokens in response to the reference-target prompt; 2) the corresponding embeddings $\mathbf{e}_{<Seg>} \in \mathbb{R}^d$ are passed through a lightweight MLP to regress a bounding box $\hat{\mathbf{b}} = (x, y, w, h) \in \mathbb{R}^4$:

$$\hat{\mathbf{b}} = \text{MLP}(\mathbf{e}_{<Seg>}). \tag{3}$$

To further improve labeling accuracy, we extend the VLM's output to generate *all* bounding boxes corresponding to the set of anatomical labels in a single forward pass. Instead of predicting each label independently, we formulate labeling as a global matching problem: predicted boxes are jointly assigned to categories using Optimal Transport (Cuturi, 2013). This design leverages spatial relationships across labels (i.e.,the relative positions between organs)as a soft constraint during assignment. Formally, let $\{\hat{\mathbf{b}}_i\}_{i=1}^N$ denote the predicted boxes and $\{\mathbf{p}_j\}_{j=1}^N$ be the label prototypes (e.g., expected positions or embeddings derived from the reference image). We construct a cost matrix $C \in \mathbb{R}^{N \times N}$ where $C_{ij}$ represents the spatial or semantic distance between $\hat{\mathbf{b}}_i$ and $\mathbf{p}_j$. The optimal transport plan $\pi^* \in \mathbb{R}^{N \times N}$ is then obtained by solving:

$$\pi^* = \arg\min_{\pi \in \Pi} \sum_{i=1}^N \sum_{j=1}^N \pi_{ij} \cdot C_{ij}, \quad \text{s.t. } \pi\mathbf{1} = \mu, \ \pi^\top\mathbf{1} = \nu, \tag{4}$$

where $\Pi$ is the set of doubly stochastic matrices (or relaxed transport plans), and $\mu, \nu$ are marginal distributions (uniform in our case). Once the transport plan $\pi^*$ determines the optimal label-to-box assignment, we apply a refinement mechanism: for unmatched or low-confidence matches (e.g., those with large cost or low attention scores), we trigger a fallback step that re-generates candidate boxes within the unassigned region. This adaptive recovery strategy improves robustness in hard cases such as small, overlapping, or occluded targets.

Training proceeds in two steps: first, the MLP is optimized with the VLM frozen; then, end-to-end fine-tuning is performed with GRPO, guided by two manually designed reward functions:

$$R = \lambda_{\text{det}} \cdot \text{AP}(\hat{\mathbf{b}}, \mathbf{b}) + \lambda_{\text{fmt}} \cdot \mathbb{1}\{\text{output format is valid}\}. \tag{5}$$

Table 2: Labeling Accuracy across Methods. Top: ID accuracy across methods. Bottom: OOD accuracy across methods. VLM+SAM2 refers to the approach proposed in Section 3.3.

| Dataset | SFT-VQA(Baseline) | RL-VQA | RL-Bbox | VLM+SAM2 |
|---|---|---|---|---|
| *In-Distribution (ID): labeling accuracy (best score)* | | | | |
| RAOS (Whole Body CT) | 64.11% | 74.68% | 78.16% | 89.38% |
| Arcade (Vessel X-Ray) | 53.92% | 64.37% | 41.09% | 81.62% |
| *Out-of-Distribution (OOD): labeling accuracy (best score)* | | | | |
| LERA (Bone X-Ray) | 30.41% | 54.57% | 55.92% | 61.87% |
| CAMUS (Heart Ultrasound) | 40.29% | 88.33% | 55.06% | 95.41% |

**Experiment Settings** For the bbox variant, we jointly train on RAOS-CT and Arcade-X-Ray, where Arcade's challenging vessel and device configurations encourage learning non-trivial spatial layouts. As in Section 3.1, the model receives a labeled reference image, an unlabeled target image, and a reference-based instruction, and must output, in a fixed format, the coordinates of the bounding box in the reference image that corresponds to the queried region in the target (see Appendix A.3.3 for details). We then test generalization on two out-of-distribution datasets, LERA-X-Ray and CAMUS-Ultrasound, under the same reference-based VQA protocol, omitting CT/MRI datasets that are too similar to RAOS-CT. Since our aim is to probe anatomical understanding rather than low-level detection heuristics, we report labeling accuracy of the predicted bounding boxes as the primary metric.

**Experiment Results** As shown in Tab. 2. On the RAOS dataset, incorporating global bounding-box matching improves labeling accuracy, from 74.68% to 78.16%, indicating that spatial grounding via box-level alignment helps guide more accurate label assignment. However, this improvement does not generalize well to structurally complex datasets such as Arcade, where the bounding-box–based method yields only 41.09% accuracy. The drop highlights a key limitation: elongated or topology-fragile structures like vessels are poorly captured by rectangular boxes, which constrains model generalization across anatomical types. Importantly, these results also suggest that reinforcement learning enables the VLM to attend to the correct regions, yet the accuracy remains limited by the coarse nature of bounding boxes, motivating the adoption of finer-grained strategies such as segmentation-based matching for improved precision in reference-based labeling tasks.

## 3.3 INTEGRATION WITH SEMANTICALLY-RICH MODELS

VQA- and bbox-style prompting methods perform poorly when anatomical targets are scattered, overlapping, or thin and elongated, as bounding boxes provide only coarse localization. In addition, the autoregressive nature of language modeling limits detail preservation, hindering accurate region prediction in cluttered or ambiguous scenes.

While VLMs provide strong global reasoning and instruction-following abilities, they lack explicit memory mechanisms for fine-grained semantic recall. In contrast, SAM2 introduces a learnable Memory Bank (Ravi et al., 2024) architecture, where each slot encodes semantic and spatial cues from reference masks (Zhao et al., 2025). These embeddings serve as long-term anchors, supporting accurate segmentation even under ambiguous or noisy conditions. Therefore, to combine the complementary strengths of both modules, we design a fusion interface where the VLM guides the attention toward relevant regions via language reasoning and spatial reference, while SAM2 executes the segmentation grounded on learned semantic memory.

**Methods** Specifically, as shown in Fig. 2, the VLM generates one or more `<Seg>` tokens, whose embeddings $\{\mathbf{h}_i^{\texttt{<Seg>}}\}_{i=1}^K$ encode the linguistic-semantic information of the target region. These embeddings are then projected into the space of SAM2's memory slots via a shared MLP:

$$\mathbf{q}_i = \text{MLP}(\mathbf{h}_i^{\texttt{<Seg>}}), \quad \mathbf{q}_i \in \mathbb{R}^d, \tag{6}$$

where $d$ is the dimensionality of SAM2's memory bank. Next, during segmentation, these projected queries $\mathbf{q}_i$ are used to retrieve relevant memory slots $\{\mathbf{m}_j\}$ from the bank via dot-product attention:

$$\alpha_{ij} = \frac{\exp(\mathbf{q}_i^\top \mathbf{m}_j)}{\sum_k \exp(\mathbf{q}_i^\top \mathbf{m}_k)}, \quad \mathbf{z}_i = \sum_j \alpha_{ij} \cdot \mathbf{m}_j, \tag{7}$$

where $\mathbf{z}_i$ is the fused representation fed into SAM2's decoder to generate the final segmentation mask.

The VLM is initialized with the weights of RL-VQA (Sec. 3.1). In the SFT phase, we jointly train the embeddings of the `<Seg>` tokens generated by the VLM, the MLP projection layer, and the SAM2 decoder under segmentation supervision, while freezing the other VLM weights. The loss is a weighted sum of Dice (Sudre et al., 2017) and binary cross-entropy (BCE) (Jadon, 2020) losses. In the RL phase, we unfreeze the VLM and optimize it with GRPO. The reward directly reflects segmentation quality, defined as a weighted sum of Dice and BCE losses.

Table 3: Quantitative analysis of segmentation performance (Dice / gIoU; higher is better). SAM2/MedSAM2-Memory denotes using the original SAM2/MedSAM2 weights while providing the reference image as the memory input. SAM2/MedSAM2-Memory-Ref-SFT follows the same strategy but further applies SFT. "Our Method" corresponds to the VLM+SAM2 approach in Figure 2c.

| Method (Dice / gIoU) | In-Distribution (ID) | | Out-of-Distribution (OOD) | |
|---|---|---|---|---|
| | **Arcade** | **RAOS** | **CAMUS** | **LERA** |
| SAM2-Memory | 0.0346 / 0.0211 | 0.1084 / 0.0573 | 0.2592 / 0.1389 | 0.1493 / 0.0807 |
| MedSAM2-Memory (Ma et al., 2025) | 0.0674 / 0.0486 | 0.1807 / 0.0925 | 0.3522 / 0.2101 | 0.1814 / 0.0996 |
| SAM2-Memory-Ref-SFT | 0.1914 / 0.1149 | 0.2509 / 0.1434 | 0.3384 / 0.2037 | 0.1843 / 0.1015 |
| MedSAM2-Memory-Ref-SFT | 0.2435 / 0.1852 | 0.2965 / 0.2094 | 0.4290 / 0.2772 | 0.2603 / 0.1723 |
| LISA-SFT (Lai et al., 2024) | 0.1278 / 0.0794 | 0.2070 / 0.1130 | 0.2778 / 0.1481 | 0.1578 / 0.0785 |
| SegZero-SFT (Liu et al., 2025b) | 0.1326 / 0.0821 | 0.2145 / 0.1318 | 0.2580 / 0.1260 | 0.1799 / 0.0930 |
| Our Method | 0.6754 / 0.5099 | 0.7151 / 0.4320 | 0.7503 / 0.5133 | 0.7010 / 0.5396 |

**Experiment Settings** For the segmentation variant, we train on RAOS-CT and Arcade-X-Ray, and evaluate OOD performance on LERA-X-Ray and CAMUS-Ultrasound. Each training sample consists of a paired reference image and target image: the reference image is fully annotated, so a corresponding organ mask is available for every reference. The VLM configuration follows the settings in Section 3.1 and 3.2. For the SAM2 component, the reference image and its mask are encoded as memory tokens, and SAM2 uses this memory together with the VLM-provided conditioning on the target image to generate a segmentation mask. As in previous sections, our primary metric is labeling accuracy, while segmentation metrics are additionally reported to compare against traditional baselines and to verify the effectiveness of the proposed training scheme. (Details can be found in Appenidx A.3.4)

**Experiment Results** As shown in Tab. 3, Our Method (VLM+SAM2), although fine-tuned only on RAOS and Arcade, consistently outperforms all SAM2/MedSAM2-based baselines as well as LISA-SFT and SegZero-SFT on both ID (Arcade, RAOS) and OOD (CAMUS, LERA) datasets. Compared to the strongest SAM2 family baseline, MedSAM2-Memory-Ref-SFT, our approach improves the average Dice from 0.31 to 0.71 and the average gIoU from 0.21 to 0.50 across the four datasets, indicating substantially more reliable and fine-grained anatomical masks.

Interestingly, MedSAM2 variants, despite being pretrained on large-scale medical data, do not close this gap: when multiple candidate structures exhibit similar shapes or lack distinctive semantic cues, MedSAM2 alone cannot reliably select the correct target from the memory, leading to low Dice/gIoU. Likewise, LISA-SFT and SegZero-SFT, which possess stronger reasoning capabilities, achieve reasonable performance on ID data after SFT but degrade sharply on OOD data. This suggests that in our setting SFT largely encourages memorization of specific training images rather

than true spatial generalization: anatomical correspondence in complex clinical scenarios cannot be solved by textual reasoning alone. In contrast, explicitly combining language-guided spatial reasoning with a segmentation foundation model enables robust anatomical grounding that transfers across modalities and datasets.

In addition, to better connect with the previous tasks in Secs. 3.1 and 3.2, we report the labeling outcomes side-by-side in Tab. 2. On ID datasets, our method achieves the best maximum labeling accuracy—89.38% on RAOS and 81.62% on Arcade, exceeding RL-VQA and RL-Bbox by large margins. More importantly, Tab. 2 demonstrates strong OOD transfer with 61.87% on LERA and 95.41% on CAMUS using the same model.

Taken together, the evidence shows that relative spatial understanding is preserved after co-training with SAM2 integration and anatomical structure localization is enhanced by SAM2's granular, semantically rich features and memory mechanism. More importantly, this capability generalizes beyond the training distribution, enabling accurate structure segmentation without large-scale per-dataset fine-tuning.

## 4 ABLATION STUDY

We conduct an ablation study to examine the effect of different VLM initialization strategies within our SAM2-enhanced pipeline, as summarized in Tab. 2. Three variants are compared: (1) Vanilla, where the VLM is directly initialized from Qwen2.5VL-7B without any task-specific tuning; (2) SFT-VQA, where the VLM is initialized from a Qwen2.5VL-7B supervisedly fine-tuned on the VQA task (Sec. 3.1); and (3) RL-VQA (used in VLM+SAM2), where the VLM is initialized from the model optimized via reinforcement learning on the VQA task.

Initialization with SFT-VQA significantly improves performance on ID datasets. On RAOS and Arcade, it improves over the instruct-only baseline by large margins. However, this improvement does not generalize well to OOD datasets. For instance, while the SFT model achieves 76.36% on CAMUS, its performance on LERA remains poor (33.95%), suggesting it may overfit to dataset-specific patterns. In contrast, initialization with RL-VQA consistently improves both ID and OOD performance. On LERA, it improves labeling accuracy to 61.87%, and similarly boosts CAMUS accuracy to 95.41%. These results indicate that reinforcement fine-tuning with the end-task reward not only leads to better generalization in the trained VQA task, but also improves OOD generalization when integrated with SAM2.

We also conduct an ablation study on the quality of the reference images, with results reported in Table A2.

Table 4: Ablation on VLM initialization in the SAM2-enhanced pipeline. Top: maximum (best) labeling accuracy across methods on ID datasets. Bottom: accuracy on OOD datasets using *Our Method*.

| Dataset | Vanilla | SFT-VQA | RL-VQA |
|---|---|---|---|
| *In-Distribution (ID): labeling accuracy (best score)* | | | |
| RAOS (Whole Body CT) | 18.23% | 85.07% | 89.38% |
| Arcade (Vessel X-Ray) | 15.41% | 76.93% | 81.62% |
| *Out-of-Distribution (OOD): labeling accuracy (best score)* | | | |
| LERA (Bone X-Ray) | 30.12% | 33.95% | 61.87% |
| CAMUS (Heart Ultrasound) | 69.18% | 76.36% | 95.41% |

## 5 CONCLUSION AND DISCUSSION

We introduced RAU, a reference-based anatomical understanding framework that uses a VLM for spatial reasoning and region identification in medical images. Trained on a moderately sized dataset with VQA and/or bounding-box prediction, the VLM learns to localize anatomy via relative spatial relationships between a labeled reference and an unlabeled target. Combined with SAM2's fine-grained segmentation capability, RAU lifts this reasoning from region-level to pixel-level localization. Experiments across multiple modalities and anatomical targets—on both in-distribution and out-of-distribution datasets—show that RAU is particularly effective in challenging settings such as vessel segment labeling and ultrasound interpretation.

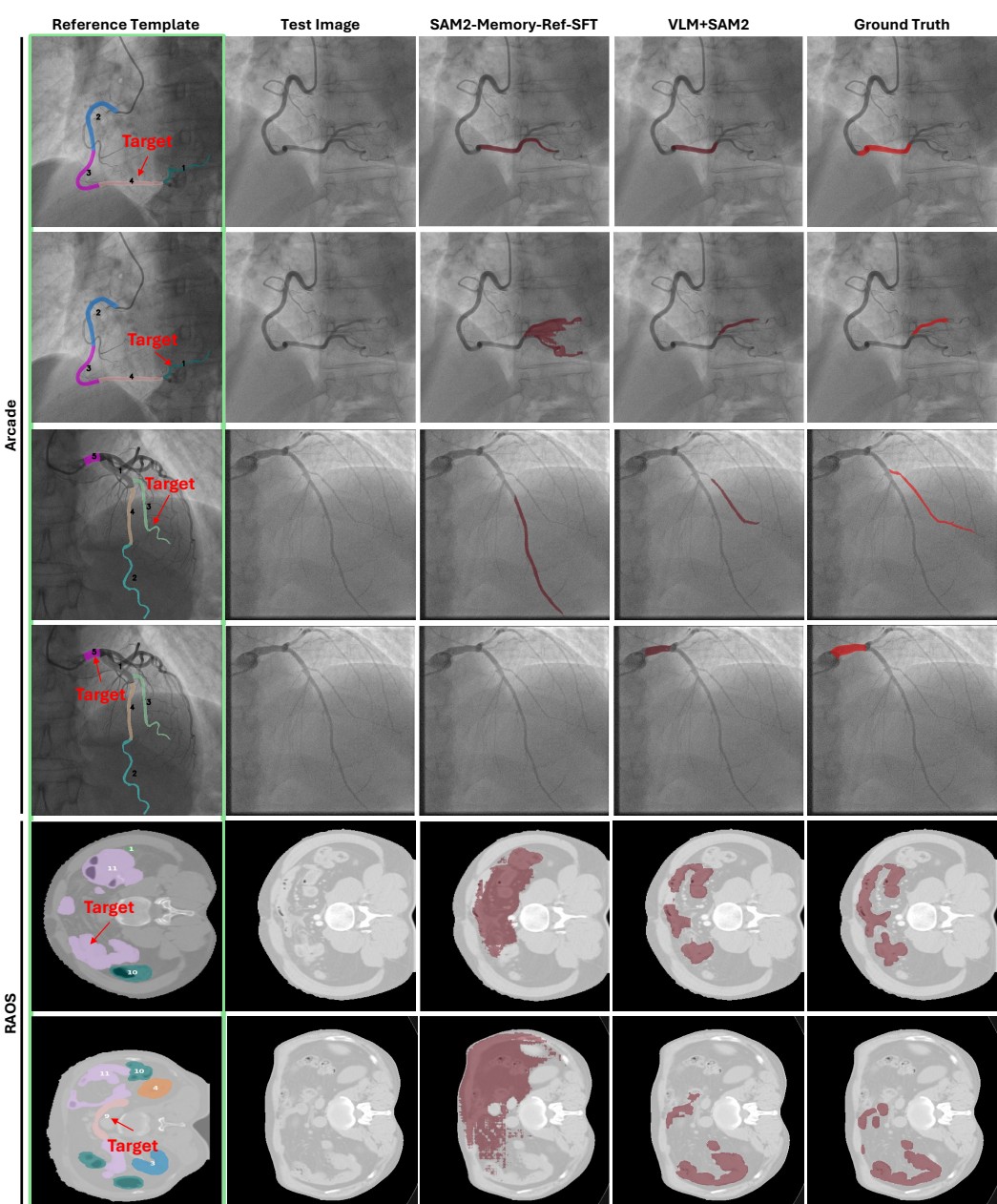

Figure 3: In-distribution qualitative results (RAOS & Arcade). Columns: reference template, test image, SAM2-SFT w/ Memory (fine-tuned on the same memory for both datasets), VLM-SAM2, and GT. Our VLM-guided SAM2 yields tighter boundaries, better continuity on elongated/fragmented vessels, and fewer leaks/misses under mild target–atlas misalignment—resulting in visibly more accurate masks than the SAM2-SFT baseline.

Looking ahead, we see several promising directions: (i) incorporating structural priors (e.g., part–whole hierarchies or organ trees) to further enforce anatomical consistency; (ii) developing adaptive memory mechanisms to better handle viewpoint and shape variations between reference and target; and (iii) extending RAU to temporal or 3D volumes, where continuity and cross-slice correspondence matter. We believe RAU provides a scalable basis for anatomical understanding under limited supervision and can serve as a building block toward clinically practical systems.

## 6 ETHICS STATEMENT

This work uses only publicly available, de-identified medical imaging datasets, thus no IRB approval was required. The method is intended as a research tool to reduce annotation burden, not for direct clinical use. We follow the ICLR Code of Ethics and caution that any deployment should involve proper clinical validation.

## 7 REPRODUCIBILITY STATEMENT

We describe architecture, training, and evaluation in detail in the paper and appendix. Datasets, pre-processing, and hyperparameters are documented, and we provide code, scripts, and model checkpoints in the supplementary materials to facilitate replication.

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

# A APPENDIX

## A.1 LLM USAGE STATEMENT

This paper does not contain any ideas, methodologies, or perspectives generated by LLMs. LLMs were only used as auxiliary tools for minor text polishing and typographical error checking. All conceptual contributions, research design, experiments, analyses, and interpretations were fully developed by the authors.

## A.2 DATASETS

We train on two primary datasets: (i) **RAOS** — a whole–body CT collection (Luo et al., 2024) containing 413 real patient scans with ground-truth annotations for 19 organs. To construct our training corpus, we extract DINOv2 features and perform cross-patient slice matching, followed by filtering with the provided GT labels to ensure anatomical consistency. This procedure yields a large-scale dataset of approximately 450k training instances, each formatted for both VQA- and bounding-box–based tasks, enabling robust supervision across diverse anatomical regions with a particular emphasis on vascular structures. During training, we additionally apply standard data augmentation—such as random flips, scaling, and rotations—to improve robustness to acquisition and pose variations.

(ii) **Arcade** — a coronary angiography dataset (Popov et al., 2024) consisting of 1,500 vessel-tree images acquired from fluoroscopic X-ray (DSA). Each image is annotated at the branch and segment level, providing dense supervision for fine-grained vascular topology. Following the same construction pipeline as RAOS, we extract DINOv2 features, perform cross-patient slice retrieval, and filter with ground-truth vessel labels to assemble a training corpus tailored for reference-based tasks. This results in approximately 21k training instances formatted for both VQA and bounding-box prediction, making Arcade a challenging benchmark for reasoning over elongated and topology-sensitive structures.

**OOD evaluation.** To assess generalization, we further evaluate on: (iii) **LERA** — a skeletal radiograph dataset (Varma et al., 2019) collected in a retrospective, HIPAA-compliant, IRB-approved study at Stanford University Medical Center, comprising radiographic examinations from 182 patients acquired between 2003 and 2014. Each study includes X-rays of the foot, knee, ankle, or hip, yielding a total of approximately Z,000 extremity images with consistent annotations of long bones and joints. The dataset is particularly suited for evaluating symmetry reasoning (e.g., left–right limb correspondence) and robustness to viewpoint variation, making it a valuable testbed for probing generalization across skeletal structures. LERA set comprises 87 images, each paired with a corresponding reference image and annotated with 7–14 segmentation labels.

(iv) **CAMUS** — a cardiac ultrasound benchmark (Leclerc et al., 2019) comprising apical two-chamber and four-chamber view sequences acquired from 500 patients. The dataset contains approximately 500 annotated videos covering end-diastolic and end-systolic phases, with expert delineations of the left ventricle, myocardium, and left atrium. Its pronounced modality gap relative to CT and DSA makes CAMUS a challenging testbed for cross-modality generalization, enabling evaluation of both structural transferability and robustness in ultrasound interpretation. The CAMUS dataset contains 900 images with corresponding reference images, each annotated with 3 segmentation labels.

**Retrieval-augmented dataset construction.** To enable retrieval-augmented training, we precompute a similarity graph over the training set. Concretely, we collect all masked training images and extract a global visual descriptor for each image using a frozen DINOv2 ViT-S/14 encoder (Oquab et al., 2023). Each image is resized to $224 \times 224$, center-cropped, and normalized with ImageNet statistics before being forwarded through DINOv2; we use the final feature vector as the image embedding. We then build a FAISS (Douze et al., 2024) over these embeddings and, for every image, retrieve its top-$K$ nearest neighbors ($K = 5$) in the embedding space. The resulting retrieval dictionary is stored as a JSON file, mapping each image ID to a list of $K$ integer IDs corresponding to its most similar reference images. This retrieval index is later used to construct RAG-style training samples, where each target image is paired with a small set of visually similar reference images as additional context.

## A.3 EXPERIMENT DETAILS

### A.3.1 MODEL SETUP AND HYPERPARAMETERS

**Implementation Details.** Model training was conducted on 8 NVIDIA A100 GPUs (80 GB memory each), leveraging data parallelism to fully utilize the available GPU memory and computational resources. The configuration of our models are as follows:

| Parameter | Value |
|---|---|
| Base VLM Model | Qwen2.5-VL-7B-Instruct |
| Segmentation Backbone | SAM-2 (Hiera-Large) |
| Epochs | 5 |
| Batch size | 1 |
| Learning rate / Optimizer | $2 \times 10^{-4}$ (AdamW) |
| Precision | bfloat16 |
| Quantization (Qwen) | 4-bit |
| Scheduler | Cosine decay with 3% warmup steps |
| Loss function | $0.7\,\mathrm{BCE} + 0.3\,\mathrm{Dice}; +\,\mathrm{CE}$ for text branch |
| Input dimension | 3584 |
| Output dimension | 256 |
| LoRA rank ($r$) | 8 |
| LoRA alpha | 16 |
| LoRA dropout | 0.01 |

Table A1: Hyperparameter Settings Used in Model Training

### A.3.2 VQA TASK

For the VQA variant, we fine-tune Qwen2.5-VL-7B-Instruct with a PEFT/LoRA setup in a multi-image setting. Each training sample consists of a *pair* of images (a labeled reference and an unlabeled target) and a short conversation. The data are stored in JSONL format with an "image" field containing a list of image paths and a "conversations" field containing a two-turn dialogue: a human turn with two <image>placeholders followed by the spatial prompt, and a gpt turn containing only the supervision signal (either a numeric region index or a structured answer).

We adopt QLoRA-style fine-tuning: the base model is loaded in 4- or 8-bit NF4 quantization with gradient checkpointing enabled to reduce memory. LoRA adapters (rank $r = 8$, $\alpha = 16$, dropout 0.05) are inserted into all attention and MLP projection layers, and the vision encoder of Qwen2.5-VL is fine-tuned jointly with these adapters, while the remaining language-side weights are kept frozen. We tokenize the full conversation and then mask all tokens corresponding to user content, computing the cross-entropy loss only on the assistant's answer tokens; thus the model is explicitly optimized to map the multi-image prompt to the desired anatomical label while treating the rest of the transcript as context.

Unless otherwise noted, we train with a per-device batch size of 1, gradient accumulation of 16 steps, learning rate $2 \times 10^{-4}$ (AdamW), and a context length of 2048 tokens. The same LoRA configuration and data pipeline are used across RAOS-CT and Arcade-X-Ray training runs; out-of-distribution evaluations on LERA-X-Ray, CAMUS-Ultrasound, AMOS2022-MRI, CT-Liver, and CT-Lung reuse the fine-tuned Qwen2.5-VL model without any additional adaptation.

### A.3.3 BBOX PREDICTION TASK

Each training sample consists of a labeled reference image, an unlabeled target image, and an instruction that specifies which structure in the target should be found. The VLM receives both images and the instruction and is required to output a structured bounding box in the reference coordinate system, parameterized as $(x_{\min}, y_{\min}, x_{\max}, y_{\max})$, that corresponds to the queried anatomy. The model configuration (backbone, tokenizer, and multi-image prompting strategy) is kept consistent with the VQA setup described in A.3.2, differing only in the bounding-box prediction head and loss design.

We adopt an adaptive reward scheme for object localization, where the bounding-box reward is binary over IoU. Training begins with AP@50 to provide dense feedback under a relatively loose overlap criterion, and the IoU threshold is then gradually increased in a curriculum manner (e.g., extending to AP@[50:5:95]) to encourage progressively more accurate and spatially grounded predictions.

### A.3.4 VLM COMBINED WITH SAM2 TASK

For all Qwen2.5-VL + SAM2 experiments, we use Qwen/Qwen2.5-VL-7B-Instruct as the VLM backbone and sam2-hiera-large as the segmentation backbone. The joint model is optimized end-to-end with a batch size of 1, learning rate $2 \times 10^{-4}$, AdamW optimizer, and a cosine learning-rate schedule with a short warm-up phase. We adopt a QLoRA-style regime with 4-bit quantization and bfloat16 compute, fine-tuning the SAM2 mask decoder (via low-rank adapters), a lightweight segmentation head on top of the VLM features, and the embedding/LM-head parameters of Qwen2.5-VL, while keeping the remaining language-side weights frozen. The supervision signal combines a pixel-wise binary cross-entropy term and a soft Dice loss (weighted $0.7/0.3$), and, when the language branch is enabled, an additional cross-entropy loss on a dedicated <SEG> token that encourages the VLM to explicitly emit the target anatomy in its output.

We jointly train the full language-guided Qwen2.5-VL + SAM2 model on the combined Arcade-X-Ray and RAOS-CT training splits, using the same configuration (10 epochs on Arcade-style data and 3 epochs on RAOS-style data) and shared hyperparameters. Throughout training, we periodically visualize predicted masks on held-out training examples from both datasets to monitor qualitative behavior.

For comparison, we also train a SAM2-only baseline (SAM2-SFT) to quantify the benefit of adding the VLM. SAM2-SFT adopts the same memory-based setup as our joint model: for each training sample, the reference image and its ground-truth mask are encoded as memory tokens by the SAM2 backbone, and the model is then required to predict the segmentation mask on the target image conditioned on this memory. We fine-tune SAM2 using the same LoRA strategy as above, attaching low-rank adapters only to the mask decoder while keeping the image encoder frozen. The training objective and hyperparameters mirror those used for the joint Qwen2.5-VL + SAM2 model, ensuring that any performance difference can be attributed to the presence or absence of VLM-guided anatomical reasoning rather than optimization or architecture confounds.

Table A2: Labeling accuracy stratified by LPIPS Distance (Zhang et al., 2018) between reference image and target image.

| Dataset | LPIPS Distance | | |
|---|---|---|---|
| | 0–0.4 | 0.4–0.6 | 0.6–1.0 |
| RAOS | 92.49% | 87.73% | 61.39% |
| Arcade | 85.92% | 80.14% | 56.10% |

### A.4 ABOUT REFERENCE IMAGE QUALITY

To quantify how the quality of the reference image affects performance, we measure the LPIPS distance (Zhang et al., 2018) between the reference and target images and stratify test cases into three similarity bins. Table A2 reports labeling accuracy for RAOS and Arcade under LPIPS ranges of 0–0.4, 0.4–0.6, and 0.6–1.0. The results show that accuracy remains high for closely and moderately matched pairs (over 90%/85% for RAOS/Arcade when LPIPS $\leq 0.4$ and only a modest drop for 0.4–0.6), and degrades more noticeably only when the reference and target become substantially dissimilar (0.6–1.0). Representative failure cases for highly mismatched pairs are visualized in Figure A6.

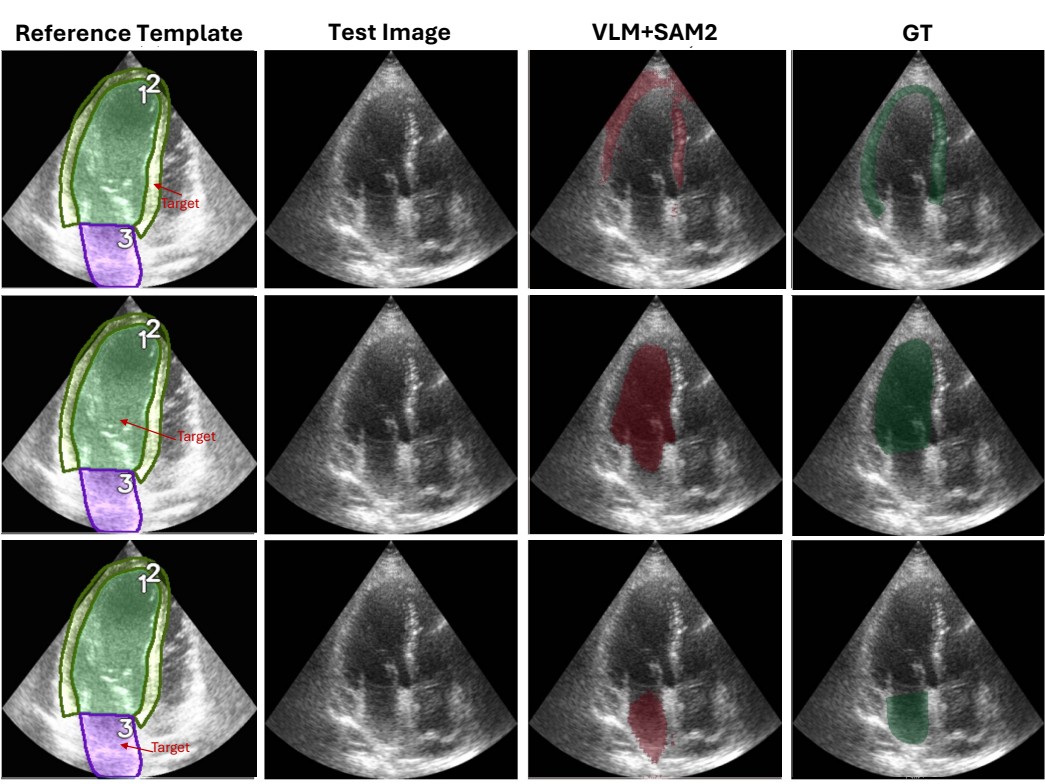

Figure A1: OOD qualitative results—anatomy/domain shift (CAMUS, echocardiography). Panels show target test images alongside their reference, our predictions, and ground truth (when available). Under severe elongation, thin branching, fragmentation, and reference–target misalignment, our VLM-guided SAM2 preserves topology, reduces leaks and misses, and adheres better to canonical segment boundaries than the SAM2-SFT w/ Memory baseline.

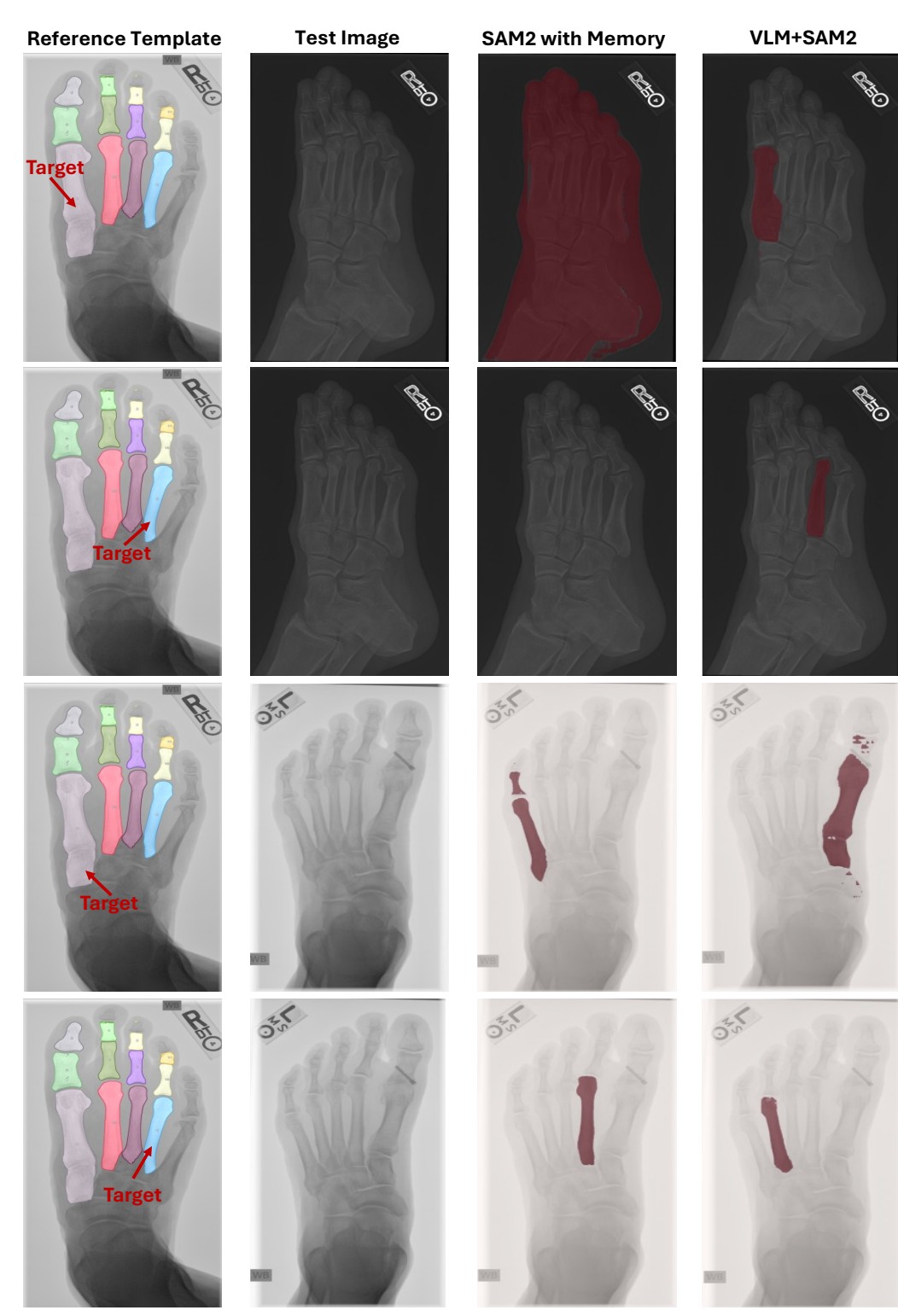

Figure A2: OOD qualitative results—modality shift (LERA). Despite ultrasound-specific artifacts (speckle, low contrast) and shape variability, VLM+SAM2 yields cleaner boundaries and more stable localization than SAM2-SFT w/ Memory (shown where applicable), demonstrating strong cross-modality generalization from reference priors.

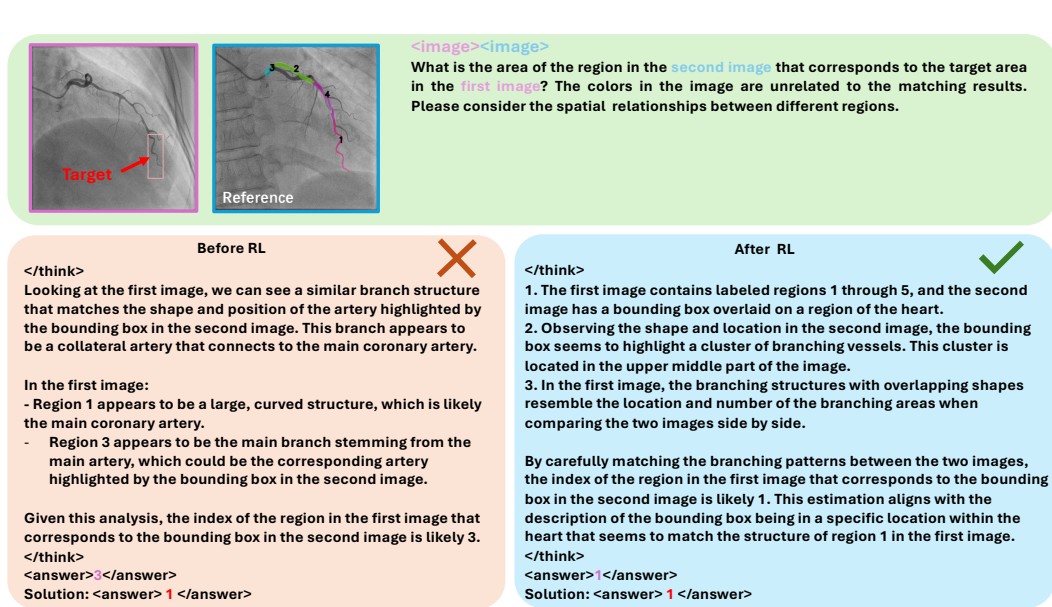

Figure A3: CoT output comparison before and after RL. Prior to RL, the VLM tends to directly identify different regions in isolation. After RL, however, the model shifts its focus toward reasoning about the relative spatial relationships among regions, which leads to higher accuracy and improved generalization.

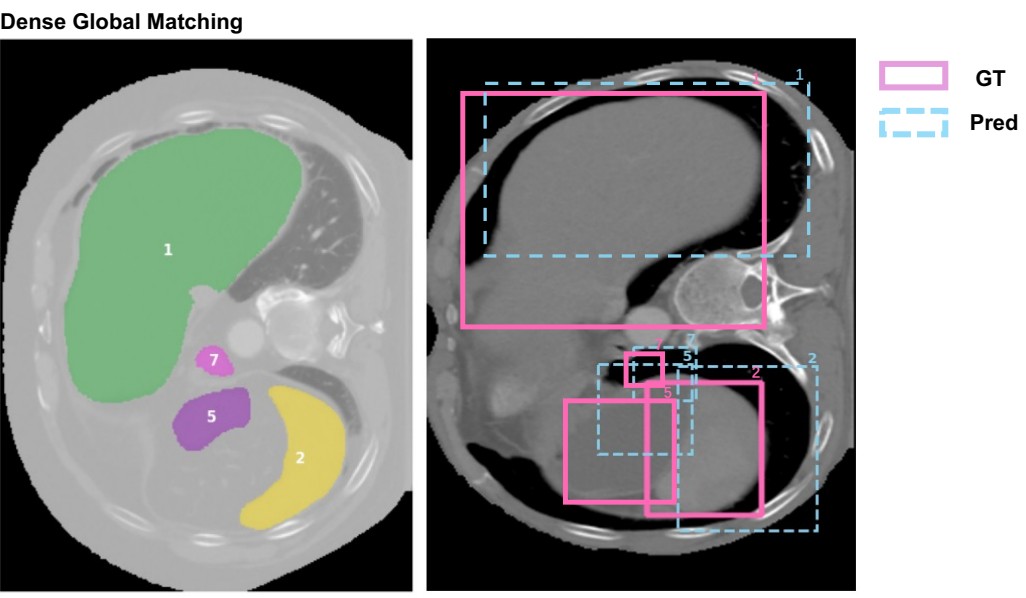

Figure A4: An example of VLM outputs with bounding boxes followed by global matching. When the number of target regions increases, global matching serves as an effective constraint to improve consistency and accuracy.

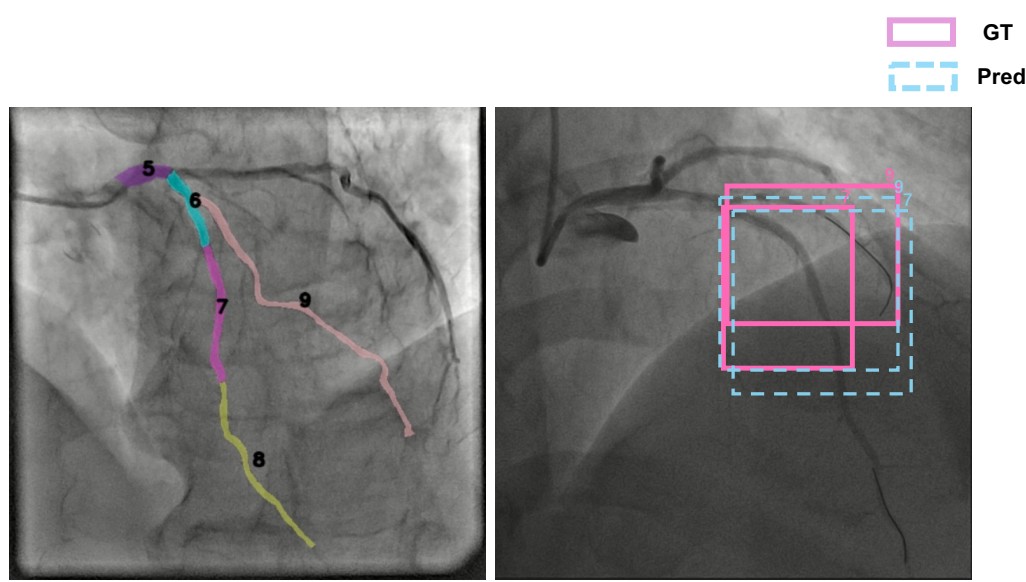

Figure A5: Arcade test-set failure cases for bounding-box prediction. In certain views, vessels are clearly visible but highly curved, causing the predicted boxes to overlap; although the model localizes the correct region, the overlapping boxes make individual targets hard to distinguish.

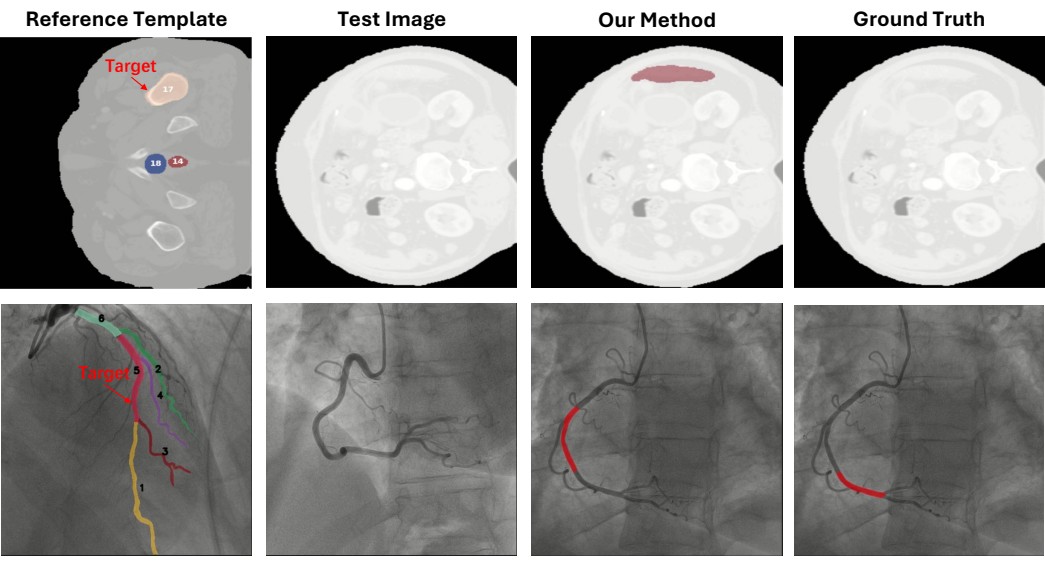

Figure A6: Failure case analysis on RAOS (CT) and Arcade (X-ray). The examples illustrate typical errors in anatomical localization and segmentation that occur when the reference image and test image differ too much, causing the model to fail to reliably identify the correct target structure.

