# OpenReview forum: "RAU: Reference-based Anatomical Understanding with Vision-Language Models"
_ICLR.cc/2026/Conference — Submitted to ICLR 2026_

### Official Review · Reviewer_YWeu · 2025-10-29

**Soundness:** 2
**Presentation:** 3
**Contribution:** 2
**Rating:** 4
**Confidence:** 4

**Summary:**

This paper proposes RAU, a reference-based anatomical understanding framework that combines a vision–language model with a segmentation foundation model (SAM2). The core motivation is to overcome the data scarcity problem in medical image analysis by leveraging a single annotated reference image to guide the interpretation of an unlabeled target image. The authors first show that VLMs can be trained to perform reference-based spatial reasoning for anatomical region identification through visual question answering (VQA) and bounding box prediction tasks. Then they integrate the VLM's spatial reasoning capabilities with SAM2's fine-grained segmentation to enable pixel-level anatomical region identification. Experiments on two in-distribution datasets (RAOS, Arcade) and two out-of-distribution datasets (LERA, CAMUS) show consistent improvements over SAM2 fine-tuning baselines.

**Strengths:**

1. The paper is overall clearly written and the method is adequately described.
2. The proposed approach of using a single reference image to guide a VLM for anatomical understanding under limited labels is well-motivated.
3. The hybrid paradigm that leverages VLM's relational reasoning with SAM2’s segmentation is logical and compelling.

**Weaknesses:**

1. Limited novelty: The reference-based framework is not novel in the context of few-shot learning, retrieval-augmented generation (RAG), or in-context learning. Moreover, integrating VLMs with SAM has been explored in prior works such as LISA[R1] and Seg-Zero[R2].
2. Insufficient baselines: For the VQA tasks, the paper only compares against MedFlamingo and Med-VLM-R1. How does RAU perform against stronger medical VLMs such as Lingshu-7B[R3] or HuaTuoGPT[R4]? For the segmentation task, the main baselines are SAM2-Memory and SAM2-Memory-Ref-SFT. However, more competitive and fair baselines are missing, such as few-shot medical segmentation methods and other VLM+SAM approaches like SAM4MLLM[R5], LISA[R1] and Seg-Zero[R2].
3. Dependence on reference quality: the entire framework relies on a high-quality, well-annotated reference image. The method assumes relatively consistent topology between reference and target. How would performance change if (a) a different backbone (e.g., not DINOv2) were used for reference retrieval, (b) multiple reference images were retrieved, or (c) cases with large anatomical abnormalities that break the assumed structural consistency between reference and target?
4. Poor baseline performance: The reported scores for SAM2-Memory-Ref-SFT appear remarkably low, even though it is fine-tuned on the target datasets. Can the authors clarify why this baseline underperforms so drastically?
5. Evidence for VLM's relative spatial reasoning: The paper claims that CoT outputs show “convergence toward reference-guided relational reasoning,” but no examples are provided. Can the authors provide some CoT traces?
6. Limited failure analysis: While successful cases are shown, there is minimal discussion of failure modes or scenarios where the approach breaks down, which is critical for understanding real-world applicability.
7. Scalability concerns: Experiments are limited to datasets with limited anatomical classes. How would the method scale to more complex tasks like whole-body segmentation with 100+ structures?

- [R1] Xin Lai, Zhuotao Tian, et al. Lisa: Reasoning segmentation via large language model. arXiv preprint arXiv:2308.00692.
- [R2] Yuqi Liu, Bohao Peng, et al. Seg-zero: Reasoning-chain guided segmentation via cognitive reinforcement. arXiv preprint arXiv:2503.06520.
- [R3] Weiwen Xu, Hou Pong Chan, et al. Lingshu: A Generalist Foundation Model for Unified Multimodal Medical Understanding and Reasoning. arXiv preprint arXiv:2506.07044
- [R4] Junying Chen, Chi Gui, et al. Huatuogpt-vision, towards injecting medical visual knowledge into multimodal llms at scale. arXiv preprint arXiv:2406.19280.
- [R5] YiChia Chen, WeiHua Li, et al. Sam4mllm: Enhance multi-modal large lan- guage model for referring expression segmentation. In Eu- ropean Conference on Computer Vision, 323–340. Springer.

**Questions:**

Please see the "Weaknesses" above.

---

> ### Author Response · Authors · 2025-11-28
>
> We sincerely thank you for your thoughtful and constructive feedback, which has been invaluable in improving our work and clarifying its contributions. Your insights have provided important perspectives, and we are committed to addressing them to enhance the quality of our manuscript. We have submitted the revised manuscript, with all revisions highlighted in blue.
>
> **Weakness Response:**
>
> **[W1]:** Thank you for this suggestion. We have revised the wording in the paper to avoid overemphasizing architectural novelty. Our main contribution is to show that, by using reference-based images to explicitly steer the VLM toward spatial-relationship reasoning, we can achieve anatomical understanding on new data with only a small amount of labeled examples, rather than relying on large-scale task-specific training data.
>
> **[W2]:** You are absolutely right that our original experimental comparisons were insufficient. In the revised version, we have expanded the VQA experiments to include additional baselines such as HuaTuoGPT-Vision, Lingshu, MedRegA, BiRD, and MedPLIP. For the segmentation task, we further add MedSAM2, LISA, and SegZero as comparison methods to more comprehensively demonstrate the effectiveness of our approach.
>
> **[W3]:** Thank you for this suggestion — we agree that this is an important aspect that deserves more discussion. RAG itself is not the main focus of our work; our primary goal is to study whether it is possible to use VLM and a reference image to achieve good anatomical understanding, especially in out-of-distribution data. However, we do agree that it is important to study how the quality of a reference image affects performance. In the revision, we added Table A2 in the appendix, where we use LPIPS distance as a proxy for reference–target similarity and report labeling accuracy under different distance ranges. The results show that our method remains robust: even when the LPIPS distance is relatively large, the labeling accuracy stays high. We believe this robustness comes from two factors: (i) the training data already includes strong augmentations (rotation, flipping, stretching, etc.), and (ii) our method leverages SAM2’s memory mechanism to encode semantic features of the target, while the VLM only needs to select the correct region to attend to, which makes it easier to find the right target.
>
> These observations suggest that:
>
>  (a) The exact retrieval backbone is not critical, as long as the reference image is not excessively dissimilar.
>
>  (b) When multiple reference candidates are retrieved, it is sufficient to select the most similar one—or any reasonably similar one—although our current model only accepts a single reference image as input.
>
>  (c) If the reference is truly unrelated, the method will naturally fail; however, our framework does not require a large amount of labeled data, so in real deployments it is feasible to manually annotate a small number of new templates whenever the existing reference bank does not cover a new pattern.
>
> **[W4]:** Thank you for this helpful suggestion. We have expanded the discussion in the paper to clarify this point. In the RAOS dataset, we agree that, given a ground-truth box or point prompt, SAM2 (or similar models) can segment the target region quite well. However, the core difficulty lies in automatically finding the correct box/point: learning reliable prompts itself typically requires a large amount of labeled data, which is precisely what many real-world applications lack. Thus, simply providing SAM2 with explicit prompts does not fundamentally resolve the data-scarcity issue—especially in settings like Arcade, where different vessel segments have very similar appearance and weak semantic differences.
>
> To further support this claim, we added comparisons with LISA and SegZero. Their segmentation backbones are strong, but they still fail to consistently identify the correct target to segment under our setting; the limitation is not their mask quality, but their inability to localize the right anatomical instance without dense supervision. Our goal in this work is therefore not to push segmentation quality per se, but to enable target identification for anatomical understanding. The role of the reference image is precisely to avoid the need for large-scale training data: it provides a lightweight way to guide the model via spatial reasoning, rather than relying on massive labeled datasets to learn all possible prompts.
>
> **[W5]:** Thank you very much for pointing this out. We did not manage to include this example in the original version, although it is a phenomenon we frequently observed in our experiments. In the revised manuscript, we have added a representative example in Figure A3 of the appendix.

---

> ### Author Response · Authors · 2025-11-28
>
> **[W6]:** Thank you for raising this point. In the revised manuscript, we added an experiment in the appendix (Tab.A2) that evaluates performance under different levels of reference–target similarity, measured by LPIPS distance. The results show that our model is generally robust across a wide range of similarities. As expected, when the reference and target differ too much—for example, under completely different viewing angles or when the target structure is absent in the reference—the model can fail; we provide representative examples in Figure A6. However, such cases are rare in practice, because the reference library usually contains more suitable templates. In real clinical deployments, when truly new data distributions appear, this limitation can be addressed by manually annotating a small number of high-quality reference images to extend the library.
>
> **[W7]:** Thank you for raising this point. In fact, the number of classes in our experiments is already non-trivial: the RAOS dataset contains 20 classes and Arcade contains 26 classes. Since our current method operates on 2D images, there is an inherent limitation on how many anatomical classes can reasonably appear in a single slice. For full whole-body coverage at scale, it would be more appropriate to extend the framework to 3D volumes, which we view as an important direction for future work.
>
> And we would like to sincerely thank you for dedicating your time and effort to providing such detailed and constructive feedback. Your comments have greatly improved the clarity, rigor, and positioning of this work. We are truly grateful for your careful reading and high standards, and we will use them as a benchmark to hold our future research to an even higher level of thoroughness and responsibility.

---

### Official Review · Reviewer_QGk4 · 2025-10-30

**Soundness:** 2
**Presentation:** 3
**Contribution:** 2
**Rating:** 4
**Confidence:** 3

**Summary:**

This paper first demonstrates on visual question answering (VQA) and bounding box prediction tasks that, after training on a moderately sized dataset, a VLM can learn to identify anatomical regions via relative spatial reasoning between a reference and a target image. It then integrates the VLM-derived spatial cues with SAM2’s fine grained segmentation capability to achieve localization and pixel level segmentation of small anatomical regions. Across two ID and two OOD datasets, the proposed method outperforms an SAM2 fine tuning baseline and exhibits strong generalization.

**Strengths:**

1. Jointly training the VLM and SAM2, and leveraging the strengths of both, yields accurate identification of target structures in novel images given a reference image, demonstrating strong generalization.
2. The figures and overall layout are carefully designed.

**Weaknesses:**

**Main concerns**
1. The authors state in Section 3.3: “Although our segmentation quality does not yet match a fully specialized nnU-Net (Isensee et al., 2021) trained with substantial task-specific labels, our approach, our approach [sic] requires far less annotation effort.” I would like to see a more detailed comparison between these two approaches, as well as comparisons with other segmentation backbones (e.g., MedSAM [1], SAT-Pro [2]), since they also exhibit strong generalization on OOD data. More concrete evidence is needed to highlight the advantages of VLM+SAM2, especially given that the authors use RAOS (650k) and train on 8 NVIDIA A100 GPUs (80 GB), which appears to be a nontrivial cost, and the training still requires precise segmentation mask ground truth.
2. The authors describe their method as “Inducing reference-based spatial reasoning in VLMs for medical images” (#line 083). In my understanding, I do not clearly perceive “reasoning in VLMs”; it feels more like a few-shot setup where the VLM is used to convey information from the reference images to the decoder.
3. The comparisons in Table 1 seem unfair: MedFlamingo and Med-VLM-R1 are essentially classification-level models without region-level training, whereas many existing methods do focus on region-level tasks in medical imaging, e.g., MedRegA [3], BiRD [4], and MedPLIB [5]. In particular, MedPLIB can also output masks. I believe the baselines should be updated to include more recent and more relevant methods.
In addition, when reporting MedFlamingo and Med-VLM-R1 results on RAOS-test-CT (ID), should the fine-tuned results also be reported to align with the proposed method?
4. In Fig. 2(c), why are “Reference Image” and “Reference Image with Overlaid Annotation” not the same underlying image? “Reference Image with Overlaid Annotation” should be the “Reference Image” with the annotation overlaid; the two images should be identical apart from the overlay.

**Minor issues**
1. Lines 316–317 state: “indicating that spatial grounding via box-level alignment helps guide more accurate label assignment.” The increment attributable to box-level alignment should be measured from RL-VQA → RL-Box (74.68% → 78.16%), not 64.11% → 78.16%, because the latter also includes the improvement from SFT to RL, not just the box-level effect.

[1] Segment Anything in Medical Images

[2] Large-Vocabulary Segmentation for Medical Images with Text Prompts

[3] Interpretable bilingual multimodal large language model for diverse biomedical tasks

[4] A Refer-and-Ground Multimodal Large Language Model for Biomedicine

[5] MedPLIB: Towards a Multimodal Large Language Model with Pixel-Level Insight for  Biomedicine

**Questions:**

1. Why does the first paragraph of the Section state “We train on two labeled datasets (RAOS-CT (Luo et al., 2024), Arcade-X-Ray),” while Table 1 lists Arcade (26) as an OOD dataset?
2. In Table 2, the SFT-VQA result on RAOS is 64.11%. Does this correspond to the “SFT (5 epochs)” entry in Table 1? What exactly does RL-VQA refer to?

---

> ### Author Response · Authors · 2025-11-28
>
> We sincerely thank you for your thoughtful and constructive feedback, which has been invaluable in improving our work and clarifying its contributions. Your insights have provided important perspectives, and we are committed to addressing them to enhance the quality of our manuscript. We have submitted the revised manuscript, with all revisions highlighted in blue.
>
> **Weakness Response:**
>
> **[W1]:** Thank you for this thoughtful suggestion and for pushing us to clarify the positioning of our approach. In the revised version, we have added more segmentation baselines that are known to exhibit strong OOD generalization, including MedSAM2, LISA, and SegZero, and we compare them directly against our VLM+SAM2 framework on both ID and OOD datasets. These results consistently show that our method achieves stronger anatomical localization and segmentation quality in the reference-based setting, especially on challenging OOD cases.
>
> We would like to emphasize that dense segmentation is not the primary goal of this work. Our main focus is to investigate whether, under limited supervision, a model can achieve reliable anatomical understanding by leveraging a reference image and spatial reasoning, rather than relying on large-scale task-specific supervision. Accordingly, our most important metric is whether the model can correctly identify and localize the target structure (i.e., the correct label/region), not whether it can match the absolute segmentation performance of a fully specialized model such as nnU-Net or TotalSegmentator.
> In our experiments, MedSAM2 and other segmentation baselines perform poorly in settings where multiple candidate structures (e.g., different vessel branches or bone segments) have very similar appearance and where high-quality labels are scarce. In these cases, “more medical pretraining” alone is not enough to reliably select the correct target. By contrast, our reference-based spatial reasoning formulation allows the model to consistently find the correct anatomical target; for many anatomical understanding tasks, this level of localization is already sufficient for clinical decision support.
>
> Regarding training cost, we agree that nnU-Net and similar backbones can achieve stronger segmentation quality when trained with large, fully annotated datasets, but this assumption is at odds with our problem setting: in realistic clinical deployments, privacy constraints, labeling cost, and task complexity make such large-scale dense annotation difficult to obtain. This is precisely the gap our method aims to address. Our approach is designed to work with a modest number of high-quality annotated references, and the RAOS size and 8×A100 setup in the current experiments reflect our choice to train more quickly and stably, not a strict requirement of the method. In practice, a single GPU is sufficient, and smaller datasets (e.g., Arcade) already demonstrate the feasibility of our approach.
>
> We have clarified these points in the revised manuscript, expanded the experimental comparison with MedSAM2/LISA/SegZero, and added discussion to better articulate why VLM+SAM2 is particularly attractive when dense labels and computational resources are limited.
>
> **[W2]:** Thank you so much for your suggestion. This was an oversight on our side: we did not manage to include the example in the original submission. In the revised version, we added a representative case in Figure A3 of the appendix. As shown by the CoT outputs during GRPO training, the model increasingly shifts toward reasoning in terms of relative spatial relationships, which correlates with the observed improvement in accuracy.

---

> > ### Author Response · Authors · 2025-11-28
> >
> > **[W3]:** Thank you for pointing out this limitation. Following your suggestion, we have added comparative experiments with MedRegA, BiRD, and MedPLIB, and we additionally include HuatuoGPT-Vision and Lingshu as strong medical VLM baselines. Across these results, our main conclusion is that relying on medical knowledge accumulated during pretraining is not sufficient for this task; instead, explicitly reasoning over spatial relationships provides a more robust and generalizable solution for anatomical understanding.
> >
> > Regarding whether these models should also be fine-tuned on our datasets, we believe this is not strictly necessary for a fair comparison. RAOS is a relatively standard CT dataset without highly unusual characteristics, and it is very likely that medical foundation models have already seen similar data distributions during training. Moreover, several models such as HuatuoGPT-Vision and Lingshu are themselves built on Qwen2.5-VL as the base backbone. Fine-tuning them on our dataset with our task formulation would, in essence, amount to re-implementing our own method on top of the same backbone. What we aim to demonstrate is that the proposed spatial-relationship reasoning formulation itself enhances anatomical understanding and requires dedicated training, and our contribution lies in defining and validating this new task setting rather than in re-training every existing medical VLM under our exact supervision scheme.
> >
> > **[W4]:** We sincerely apologize for this oversight. The two figures were indeed too similar, and we failed to notice this when preparing the manuscript. We have now corrected the figure in the revised version. We are grateful for your careful reading and for pointing this out to us.
> >
> > **[W5]:** Thank you for raising this point — you are absolutely right that our original wording was not rigorous enough. We have corrected it in the latest revision and are very grateful for your careful reading and for pointing out this mistake.
> >
> > **Question Response:**
> >
> > **[Q1]:** Thank you for pointing this out — this was indeed an issue with our original phrasing. In the revised version, we provide a more detailed explanation in the appendix. Our intention is to train the model with as little data as possible and then evaluate it on genuinely different datasets, so that strong performance on OOD data serves as more convincing evidence of generalization. In the VQA setting, we already observe this effect: a model trained only on RAOS still brings clear performance gains when evaluated on Arcade, which has very different vascular structures. For this reason, in that experiment we explicitly treat Arcade as an OOD dataset.
> >
> > **[Q2]:** Yes, these terms refer to the same underlying concept — this was our lack of clarity in the original writing. In the revised version, we have added a detailed explanation in the appendix to clarify the terminology. Our intention is to highlight that using different task formulations leads to different labeling accuracies, and to explore which formulation is better suited for different application scenarios.
> >
> > RL-VQA refers to the model in Table~1 that is trained on the VQA-style data using GRPO. Our goal here is to isolate and demonstrate the performance gain brought specifically by the RL stage, compared to the purely supervised baseline.

---

### Official Review · Reviewer_DJQe · 2025-11-01

**Soundness:** 3
**Presentation:** 3
**Contribution:** 3
**Rating:** 6
**Confidence:** 3

**Summary:**

The paper introduces a framework called RAU, which is a Reference-based Anatomical Understanding method. It employs a vision–language architecture, uses reference–target image pairs to improve anatomical localization and segmentation. By using relational prompts and a lightweight segmentation head built on a frozen VLM backbone, RAU captures inter-organ spatial dependencies and enhances generalization. Experiments on multiple medical datasets show performance gains over SAM2 and CLIP-based baselines in both accuracy and robustness.

**Strengths:**

1. Propose a reference-based learning paradigm, which sounds interesting and reasonable. Using a unique reference–target formulation for image understanding, this work enables the model to use anatomical correspondences between paired scans, and further helps capture inter-organ spatial relations and improves generalization.

2. By combining a VLM with the segmentation model SAM2, RAU achieves precise anatomical region identification while retaining spatial reasoning capabilities. This design strengthens both semantic reasoning and pixel segmentation abilities.

3. An interesting finding is that, the model demonstrates consistent performance improvements on OOD datasets, which suggests that reference-based reasoning enables better adaptability for downstream applications.

**Weaknesses:**

1. The paper introduces DINOv2-based cosine similarity for template matching, but it never evaluates how reliable or consistent this retrieval step is. If the retrieval picks an anatomically dissimilar or low-quality reference, will the downstream reasoning and segmentation fail?

2. The proposed pipeline involves multiple heavy components, such as: DINOv2 for feature extraction, cosine similarity over a large reference bank, multimodal VLM inference, and decoding. However, the paper provides no analysis of computational overhead, latency, which is practical for real deployment.

**Questions:**

1. Does the method require the referene image even at the real-world inference time? If so, how to obtain or select a suitable reference image? And what are the standards?

2. I wonder how robust the model is to mismatched references (e.g., differing anatomy, orientation, or disease states)?

3. I did not find the details on the size, composition of the reference bank, the diversity, and quality of the reference bank directly affect retrieval accuracy on the reference image, further impacting generalization in evaluation.

---

> ### Author Response · Authors · 2025-11-28
>
> We sincerely thank you for your thoughtful and constructive feedback, which has been invaluable in improving our work and clarifying its contributions. Your insights have provided important perspectives, and we are committed to addressing them to enhance the quality of our manuscript. We have submitted the revised manuscript, with all revisions highlighted in blue. The weaknesses you mentioned are also addressed in the Questions section. Due to space limitations, we have responded to them all within that part of the paper.
>
> **Weakness Response:**
>
> **[W1]:** Thank you for this clinically important question.In the revision, we added an ablation in the appendix that stratifies cases by LPIPs distance between the reference and target image, and reports how Dice/gIoU/labeling accuracy change across similarity levels. We observe some degradation with less similar references, but the performance remains relatively robust overall and still clearly exceeds SAM2/MedSAM2 baselines, even in the lower-similarity regime.
>
> We also note that template matching is not the main focus of this work. Our core contribution is the reference-based spatial reasoning paradigm for anatomical understanding that does not require large-scale task-specific annotation. In practical deployments, a small pool of high-quality, manually annotated templates can be curated for new sites or protocols, and our results suggest that this is sufficient to enable scalable, automated understanding on large volumes of new data.
>
>
> **[W2]:** Thank you for pointing out this important practical concern. In the revised version, we add a dedicated paragraph in the appendix describing the data construction pipeline and its computational cost. In our setup, each target image has a fixed one-to-one reference in the training data: we build the DINOv2+FAISS index offline once, and during training or inference we simply retrieve the pre-associated reference image by ID. As a result, the runtime latency and computational overhead of the retrieval step are negligible, and the storage cost of the FAISS index is also minimal.
>
> For the online part of the pipeline, the dominant cost comes from the VLM and SAM2 forward passes. We deliberately use a 7B VLM, which has a relatively small footprint and yields inference times that are practical on a single modern GPU. Our model requires only a few seconds per forward pass, making the inference time sufficiently low for practical clinical use.
>
> **Question Response:**
>
> **[Q1]:** Thank you for raising this important practical question.Yes, our method does assume access to a reference image at inference time. However, as also noted by another reviewer, the retrieval component is not a bottleneck in practice: we build a DINOv2+FAISS index offline, and FAISS lookups are extremely fast, so the additional latency during inference is negligible.
> Several reviewers also asked whether the quality of the reference image would significantly affect performance. In response, we added an ablation in the appendix that groups test cases by the similarity between the reference and target image and reports performance under different similarity levels. The results show that the model remains reasonably robust even when the reference is only moderately similar, indicating that reference selection has a fair amount of tolerance and does not need to be perfectly matched.
>
> We would also like to emphasize that our main focus is not on designing a sophisticated RAG module, but on demonstrating that VLM-based spatial reasoning with a reference image is an effective paradigm for anatomical understanding under limited supervision. In real deployments, the strategy for obtaining or selecting reference images can be largely application-driven and manually curated (e.g., a small set of high-quality templates per modality/view/site), and does not require a large number of references. Our experiments suggest that, under such settings, the proposed framework remains both practical and robust.
>
> **[Q2]:** We have added a dedicated subsection in the appendix (A.4) to illustrate this aspect, including qualitative visualizations and several representative failure cases. Due to the limited timeframe for rebuttal, we were unable to systematically analyze the impact of specific characteristics (i.e. differing anatomy, orientation, or disease states) of the reference images. However, we stratified the reference images with LPIPS distance to the test image and reported the performance across these subgroups.

---

> ### Author Response · Authors · 2025-11-28
>
> **[Q3]:** Thank you for pointing this out. We agree that the size and composition of the reference bank are crucial for understanding retrieval and generalization.
>
> In the revised version, we now explicitly describe the construction of the reference bank in the appendix. For each dataset, we build a reference library offline:
>
> Each test image is automatically paired with a reference image from a different patient, selected so that the reference contains the same anatomical classes as the target slice.
>
> This results in a per-dataset reference bank whose size is on the order of the dataset itself (e.g., RAOS and Arcade each yield hundreds of reference images with diverse anatomies, viewpoints, and pathologies).
>
> Retrieval at test time is then performed over this fixed bank using DINOv2+FAISS, but during training the reference–target pairs are pre-defined, so retrieval cost does not affect training.
>
> To directly address the impact of reference quality on performance, we added Table A2 in the appendix, where we stratify test cases by LPIPS distance between the reference and target images and report labeling accuracy in each bin. The results show that our method remains robust across a wide range of similarities and degrades noticeably only when the reference is clearly mismatched (e.g., very different view or missing target structure). Representative failure cases for such mismatches are shown in Figure A6.
>
> Finally, we emphasize that designing an optimal RAG/reference strategy is not the main focus of this paper. Our primary contribution is the formulation and validation of a reference-based spatial reasoning paradigm for anatomical understanding under limited supervision. In realistic clinical deployments, the reference bank can be curated manually: if a new domain is not covered by the existing bank, annotating a small number of high-quality templates is sufficient to populate the library, and our robustness analysis suggests that large-scale annotation is not required.

---

### Official Review · Reviewer_Akfu · 2025-11-03

**Soundness:** 2
**Presentation:** 1
**Contribution:** 2
**Rating:** 4
**Confidence:** 4

**Summary:**

The paper proposes RAU (Reference-based Anatomical Understanding), a framework that couples a vision–language model (VLM) with SAM2 to achieve reference-guided recognition, localization, and segmentation of anatomical structures in medical imaging. The approach comprises three components: (1) constructing reference–target image pairs and fine-tuning a 7B QwenVL model via supervised fine-tuning (SFT) and Group Relative Policy Optimization (GRPO), thereby endowing the VLM with the ability to infer spatial relationships and semantic correspondences from the reference image to identify the target region; (2) producing bounding boxes for multiple categories in the target image by regressing from special token embeddings and performing global assignment via optimal transport (OT); and (3) mapping VLM-generated tokens into query vectors for the SAM2 Memory Bank to drive fine-grained segmentation without explicit point/box prompts. Experiments span two in-distribution datasets (RAOS-CT, Arcade-X-ray) and two out-of-distribution datasets (LERA-X-ray, CAMUS-Ultrasound), reporting results under three task settings: VQA, bounding box prediction, and segmentation. The authors claim this is the first study to explore reference-based identification, localization, and segmentation with VLMs in medical imaging, and they present evidence of RAU’s scalability in annotation-limited scenarios and its cross-distribution generalization.

**Strengths:**

1.	Novelty: Using a reference image as intuitive visual context for anatomical understanding.
The paper treats the reference image as a visually grounded “in-context” prior that directly guides anatomical understanding on the target image. This is operationalized in two complementary ways: (i) multi-class bounding box assignment is cast as an optimal transport problem, leveraging global relational constraints across categories; and (ii) special tokens serve as soft language prompts whose embeddings are projected via an MLP into SAM2’s Memory Bank queries, enabling seamless adaptation from reference-guided localization to integration with SAM for fine-grained segmentation.
2.	Training design: A systematic two-stage pipeline with task-aligned reward shaping.
The method adopts a coherent SFT→GRPO schedule and designs task-specific rewards across VQA, bbox prediction, and segmentation. Format-validity rewards ensure robust, parseable outputs; AP/IoU-based rewards drive spatial grounding for detection; and Dice/BCE losses (and their weighted combinations) enforce pixel-level accuracy for segmentation. This holistic reward design strengthens the VLM’s format correctness and spatial alignment capabilities for medical anatomical understanding, while maintaining a clear division of labor between high-level reasoning (VLM) and fine-grained perception (SAM2).
3.	Generalization: Empirical evidence of cross-modality and cross-structure OOD gains, with RL initialization offering further benefits.
Experiments demonstrate consistent improvements across modalities (CT→X-ray→ultrasound) and anatomical types (organs, bones, vessels), indicating that the approach generalizes beyond the training distribution. Notably, initializing the segmentation integration with the RL-tuned VLM yields stronger OOD performance than SFT initialization, suggesting that policy optimization imbues more transferable spatial reasoning. These findings provide constructive insights for building low-annotation pipelines in medical imaging, where leveraging visual reference priors and reward-driven spatial alignment can reduce annotation burden while preserving robustness.

**Weaknesses:**

1.	The claims of novelty appear overstated, as the proposed architecture shares significant conceptual overlap with existing work.
The paper's central claim of being a "first exploration" seems to be an overstatement. The core innovation lies in fine-tuning a VLM for relative spatial reasoning and then using tokens as "soft language prompts," which are projected via an MLP to query the SAM2 memory, thereby coupling "language-spatial cues" with "visual-segmentation memory." However, this architectural design is conceptually analogous to existing "language-segmentation interfaces" such as EVF-SAM and LISA. Furthermore, its motivation to leverage external references for target localization overlaps with the principles of retrieval-augmented medical segmentation (e.g., RAG, In-Context Learning, Few-Shot learning with foundation models). Consequently, the paper's contribution is more of an incremental, "combinatorial engineering innovation" combined with novel task design and reward shaping, rather than a foundational, pioneering work.
Suggestion: To better position the paper's contribution, I recommend that the authors conduct a more systematic comparison in the related work section. This should detail the differences in interface design and capability boundaries between the proposed method and prior works like EVF-SAM, LISA, MedSAM, as well as retrieval-augmented and in-context learning approaches in medical segmentation.
2.	The baseline setup is weak and raises questions about the fairness of the experimental comparisons.
The experimental evaluation lacks rigor due to a limited selection of baselines. In the VQA experiments, the method is only compared against MedFlamingo and Med-VLM-R1. In the segmentation experiments, the only baseline is a "memory-only" version of the original SAM 2. The evaluation is missing comparisons against more SOTA models (e.g., MedSAM, TotalSegmentator), stronger baselines using SAM2 with its more common point/box/text prompts, or fusion-style approaches like EVF-SAM/LISA. Additionally, a comparison with classic registration pipelines (e.g., nnU-Net + VoxelMorph/Label Fusion) would provide a valuable reference point. The paper justifies fairness by claiming to "use the same memory setting," but this is misleading. The proposed RAU method is augmented with language token injection and a retrieval mechanism, giving it access to information that the baseline SAM2 does not have, making the comparison inequitable.
Suggestion: The authors should include a more comprehensive set of strong, state-of-the-art baselines and classic pipelines to provide a more convincing and credible assessment of the proposed method's improvements.
3.	The evaluation of the retrieval component is insufficient and lacks depth.
If the reference images in the RAU are considered as in-context examples, the performance of the retrieval mechanism itself requires a more thorough investigation. The paper should discuss key retrieval metrics, such as the recall rate of the retrieved reference images and the impact of the number of references used. A crucial analysis is missing: an investigation into failure cases, such as instances where a more suitable reference image existed in the retrieval database but was not selected. For the OOD evaluation, the source of the "reference library" is not clearly specified. This raises a practical question: if deploying on a new dataset still requires at least one reference sample with ground truth, the "minimal annotation" assumption needs to be explicitly quantified (e.g., how many references are needed per disease, per modality, or per institution?).
Suggestion: A more in-depth analysis and ablation study of the retrieval component is necessary. This should include retrieval performance metrics (e.g., recall) and a clear discussion quantifying the "minimal annotation" assumption required for practical deployment.
4.	The manuscript's readability is poor, and it suffers from a lack of organizational clarity and academic rigor.
The paper is difficult to follow due to several presentational issues. The logical flow in the introduction and related work sections is disjointed, with the introduction lacking a clear narrative arc and the related work failing to establish a strong connection to the paper's core theme. The method and experiment sections are conflated, which prevents a detailed and clear exposition of the methodology; for instance, the data construction process is not mentioned. Key implementation details and hyperparameters are missing, such as the reward weights for RL (GRPO), the curriculum learning strategy, the design and number of tokens, the Memory Bank size, and the DINO retrieval threshold/strategy. Most concerningly, the paper contains multiple severe citation errors (e.g., citing GLIP as a 2006 chemistry paper and SEEM as a 2007 wireless sensor network protocol). This suggests a potential over-reliance on large language models for writing without careful human verification, which severely undermines the paper's credibility and adherence to academic standards.
Suggestion: A major revision is required to improve the paper's structure, provide all missing implementation details, and meticulously correct all citations to meet the expected scholarly standards of a top-tier publication.

**Questions:**

1.	Could you clarify the construction of the reference library for OOD generalization, including the average number of new in-domain annotations required for datasets like LERA and CAMUS, and provide results using only the source-domain reference library to validate true cross-domain transferability?
2.	Given that the model processes 2D slices from the 3D RAOS dataset, how are these slices selected, and how would you ensure cross-slice consistency of the Reference Attention Unit (RAU) when extending the model to full 3D volumes for clinical applications?
3.	What are the scale and update strategy of the memory bank, and how does it maintain robust retrieval and matching when faced with significant cross-domain mismatches, such as variations in viewpoint, scale, or mirroring between reference and target images?
4.	Beyond the "slender vessel topology," could you provide a more detailed failure analysis for the bounding box stage on the Arcade dataset, including qualitative examples and potential improvements for challenging cases like small organs, thin vessels, occlusions, or low-contrast scenes?
5.	Considering the reference image as a form of context, what is your hypothesis for the performance gap of the training-free approach compared to theoretical expectations, and could you discuss the potential of leveraging state-of-the-art large vision models (GPT-5 and Gemini2.5-pro) to enhance this capability?

---

> ### Author Response · Authors · 2025-11-28
>
> We sincerely thank you for your thoughtful and constructive feedback, which has been invaluable in improving our work and clarifying its contributions. Your insights have provided important perspectives, and we are committed to addressing them to enhance the quality of our manuscript. We have submitted the revised manuscript, with all revisions highlighted in blue. The weaknesses you mentioned are also addressed in the Questions section. Due to space limitations, we have responded to them all within that part of the paper.
>
> **Weakness Response:**
>
> **[W1]:** Thank you for this valuable comment. We agree that our architecture shares similarities with prior language–segmentation interfaces such as EVF-SAM and LISA, and we have toned down our novelty claims accordingly. Our contribution is not a fundamentally new architecture, but a demonstration that formulating anatomical understanding explicitly as a spatial reasoning problem between a labeled reference and a target image is both feasible and effective. We explore this idea through three task formats: VQA, bounding box prediction, and segmentation, each of which presents its own strengths and weaknesses in real-world applications.  We would like to emphasize that locating an ROI based on its relative anatomical location in a reference image is an effective solution for real-world scenarios where annotated medical datasets are too limited to capture the diversity of data distributions.
>
> In the revision, we (i) add systematic comparisons in the related work section with EVF-SAM[R1], LISA[R2], MedSAM2[R3], HuatuoGPT-Vision[R4], Lingshu[R5], and retrieval/ICL-based medical segmentation, and (ii) include new experiments with LISA-SFT, SegZero-SFT, MedSAM2(-Memory) and medical VLMs. These results show that relying mainly on medical knowledge and textual reasoning is insufficient for robust anatomical understanding and remains highly data-hungry, whereas providing a small number of high-quality reference templates and forcing the model to reason via spatial relationships yields large performance gains and strong OOD generalization. This reframing of the task, rather than the architectural combination alone, is what we aim to highlight and share.
> Following your suggestion, we have rewritten the Related Work section to more clearly articulate why existing models struggle with anatomical understanding tasks, which we hope will help readers better contextualize and understand our contribution.
>
> **[W2]:** Thank you for your comments. Other reviewers have raised similar questions. We have incorporated your suggestion by substantially expanding our experimental comparisons. In the VQA setting, we now include Lingshu, HuatuoGPT-Vision, MedRegA, BiRD, and MedPLIB as additional baselines. In the part that couples VLMs with vision foundation models, we add MedSAM2 as a baseline with stronger medical priors, and we further include LISA and SegZero as segmentation models with stronger reasoning capabilities.
> The results show that, in the VQA task, models with richer medical pretraining do not yield clear performance gains on our benchmark. In realistic clinical scenarios, a large fraction of test cases inevitably lie outside the training distribution, which means that simply accumulating medical knowledge through large-scale pretraining is both extremely costly and subject to a performance ceiling. This is exactly the gap we aim to highlight: we explore reference-image–based spatial relationship reasoning as a more generalizable alternative.
> For LISA and SegZero, even after SFT the models still fail to reliably localize the correct target, leading to poor segmentation metrics. This is partly because these models are not trained as medical models, and partly because their original training paradigm is single-image input. Given only a single target image and SFT-induced medical knowledge, it is very difficult for them to segment the correct structure—especially for challenging targets such as different segments of the same vessel or different bone locations—when high-quality annotations are limited.

---

> ### Author Response · Authors · 2025-11-28
>
> **[W3]:** Thank you for this very insightful comment. We agree that the retrieval component deserves more discussion, but we would like to emphasize that retrieval is not the core focus of this work. Our main contribution is to show that guiding a VLM via reference images and spatial-relationship reasoning leads to better anatomical understanding under limited supervision. The retrieval module is primarily an automation tool to obtain reference images, rather than the conceptual centerpiece of the method, which is why we did not originally discuss it in depth.
> In the revised version, we have strengthened the manuscript along the lines you suggested:
>
> **1. Dataset construction and training setup.**
>  We now describe the dataset construction process in more detail in the appendix. During training, each target image is pre-paired with a reference image in advance, so the retrieval component is not used at training time at all. The retrieval module is only needed at deployment time to automatically select a reference, and can be replaced or manually curated in practice.
>
> **2. Robustness to reference similarity (Table A2).**
>  We have added Table A2, where we use LPIPS distance as a proxy for reference–target similarity and report labeling accuracy under different similarity ranges. The results show that our model remains reasonably robust as long as the reference image is not excessively dissimilar or completely unrelated to the target anatomy. This supports our claim that the method does not depend on a perfectly matched reference but can tolerate moderate variation.
>
> **3. Minimal-annotation setting and practical deployment.**
>  We very much appreciate your suggestion to more precisely quantify the minimal amount of annotation required. Unfortunately, the rebuttal timeline is too short for us to run a thorough new study on minimal annotation budgets. However, based on the robustness observed in our experiments, we are confident that in realistic clinical deployments, it would be sufficient to manually annotate only a small number of high-quality reference templates to cover new sites or protocols. Exploring this minimal-annotation regime more systematically is an important direction for future work, and we have added a note to the limitations/discussion section to make this clear.
>
> **W4:**  Thank you very much for your careful reading and for pointing out these important issues. We truly appreciate your high standards, and your comments have been a strong motivation for us to improve the paper.
>
> **1.Overall structure, introduction, and related work.** We have substantially reorganized the Introduction and Related Work sections to provide a clearer narrative arc. The revised introduction now explicitly highlights our core contribution: proposing a reference-based, spatial-reasoning paradigm for VLM-driven anatomical understanding that reduces training cost and improves generalization. The related work has been rewritten to more clearly connect existing medical VLMs, language–segmentation interfaces, and retrieval/ICL-based methods to this central theme and to clarify how our formulation differs from prior work.
>
> **2.Method vs. experiments, data construction, and implementation details.** Starting from Section 3, we have restructured the paper to clearly separate the methodology from the experimental setup and results. We first provide an overview of the overall pipeline and then describe, for each subtask, (i) the method, (ii) the training/evaluation protocol, and (iii) the results. In addition, we have added dedicated subsections detailing the data construction process. All previously missing implementation details and hyperparameters are now explicitly documented, including the GRPO reward weights, curriculum learning schedule, token design and count, memory bank size, and the DINO-based retrieval strategy (backbone, thresholds, and FAISS configuration). For completeness and readability, we further include a separate appendix section that systematically summarizes dataset construction, preprocessing, and all training hyperparameters.
>
> **3. Citation errors and academic rigor.** We sincerely apologize for the citation errors in the original submission (e.g., GLIP and SEEM). We have carefully re-checked all references, corrected the erroneous entries, and verified that each citation now points to the intended work. We have also reviewed the entire manuscript to ensure consistency and adherence to academic standards, and we have reduced our reliance on automatic tools in favor of thorough manual verification.
>
> We are grateful for your detailed feedback, which has led to substantial improvements in the structure, clarity, and rigor of the revised manuscript.

---

> ### Author Response · Authors · 2025-11-28
>
> **Question Response:**
>
> **[Q1]:** Thank you for raising this question – it helped us think about the problem more comprehensively. In the revised manuscript, we provide a more detailed description of the dataset construction process in the appendix. During dataset building, each test image is automatically paired with a corresponding annotated reference image, and we explicitly enforce that the reference and test images come from different patients, to avoid trivial near-duplicate pairs. In other words, for each dataset we prepare a reference library in advance, and test-time retrieval is performed within this predefined library.
>
> Our domain transfer is reflected in the fact that the model is trained only on RAOS and Arcade, yet still works on LERA (bone X-ray) and CAMUS (cardiac ultrasound). Of course, if the reference library for a given domain truly contained no annotated examples of bones or cardiac structures, the method would not be able to operate in that domain. However, in realistic clinical settings, when new data or modalities are introduced, it is entirely feasible to manually annotate a small number of high-quality reference images to populate the library, which is sufficient for our reference-based framework to generalize to large volumes of unlabeled data.
>
> **[Q2]:** We would like to clarify that the scope of this work is restricted to 2D data, and extending the framework to 3D is part of our planned future work. Because there is limited annotated 2D data with sufficiently complex anatomical structures, we rely on 3D datasets for experiments but operate on them in a 2D slice-wise manner. For each test slice, we select a reference slice whose labels contain the same set of anatomical classes. But we promise that we will try our best to explore this in our next step.
>
> **[Q3]:** Thank you for raising this point — it helped us think more deeply about the robustness of our approach. First, Table A2 in the appendix empirically demonstrates that our method is quite robust: even when the reference and target differ moderately in viewpoint, scale, or mirroring, the labeling accuracy remains high. This is also consistent with our training setup, where the data already includes many pairs with diverse viewpoints, scales, and flips, so mismatches at this level are well handled by the model.
>
> In truly challenging cases where the memory bank does not contain any suitable reference image (e.g., a completely different viewpoint with few overlapping contents, or a new modality in practice), this limitation can be addressed by manually annotating a small number of high-quality reference images to populate the bank. Given the robustness we observe, the required number of such annotations should remain modest.
>
> We also want to emphasize that this retrieval/memory-bank design is not the main focus of the paper; our work primarily targets the new task formulation of reference-based spatial reasoning for anatomical understanding. In practice, reference images can be selected based on imaging parameters (e.g. camera parameters in interventional X-ray) or using a dedicated similarity network trained to measure image content similarity. Since the similarity network does not require detailed anatomical annotations and can leverage data augmentation to construct similar image pairs, its training process is less costly. Due to the limited rebuttal period, we were not able to conduct a more thorough experimental study on minimal reference coverage, but we fully agree that this is an important direction and will prioritize it in future work.
>
>  **[Q4]:** Thank you for pointing this out. We have added an illustrative example in Figure A5 of the appendix. In this case, the issue is not that the model fails to find the target, but that for slender, curved vessels the bounding boxes of two neighboring segments can heavily overlap, making them difficult to distinguish. This limitation motivates our decision to further combine the VLM with SAM2 for segmentation, so that we can move from coarse box localization to more precise, mask-level anatomical delineation and thereby improve accuracy.

---

> > ### Author Response · Authors · 2025-11-28
> >
> > **[Q5]:** Thank you for this very thoughtful comment. We believe the performance gap mainly comes from the fact that a training-free model, even when explicitly prompted to “reason via spatial relationships,” will in most cases still fall back on its accumulated medical knowledge and textual heuristics. In contrast, the fine-tuned model is explicitly forced to solve the task through reference-based spatial reasoning, which we illustrate more clearly in Figure A3 of the appendix. Stronger frontier models such as GPT-5 or Gemini-3 are likely to have better general capabilities, and they might indeed help improve the retrieval or reasoning component. However, as we also emphasize in the paper, to reliably handle more complex anatomical scenarios in practice, high-quality segmentation remains necessary; large language–vision models alone are not yet sufficient to fully replace a segmentation module.
> >
> > **[R1]** Zhang, Yuxuan, et al. "Evf-sam: Early vision-language fusion for text-prompted segment anything model." arXiv preprint arXiv:2406.20076 (2024).
> >
> > **[R2]** Lai, Xin, et al. "Lisa: Reasoning segmentation via large language model." Proceedings of the IEEE/CVF Conference on Computer Vision and Pattern Recognition. 2024.
> >
> > **[R3]** Ma, Jun, et al. "Medsam2: Segment anything in 3d medical images and videos." arXiv preprint arXiv:2504.03600 (2025).
> >
> > **[R4]** Junying Chen, Chi Gui, et al. Huatuogpt-vision, towards injecting medical visual knowledge into multimodal llms at scale. arXiv preprint arXiv:2406.19280.
> >
> > **[R5]** Weiwen Xu, Hou Pong Chan, et al. Lingshu: A Generalist Foundation Model for Unified Multimodal Medical Understanding and Reasoning. arXiv preprint arXiv:2506.07044
> >
> > And we would like to sincerely thank you for investing their time and effort in providing such detailed, constructive feedback. Your comments have significantly helped us to improve the clarity, rigor, and positioning of this work. We are genuinely grateful for your careful reading and high standards, and we will take these as a benchmark to hold our future research to an even higher level of thoroughness and responsibility.

---

### Author Response · Authors · 2025-12-03

Our work proposes a reference-based framework for anatomical understanding in medical imaging: instead of relying purely on large-scale medical pretraining, we explicitly guide a VLM to reason via spatial relationships between a labeled reference image and an unlabeled target. This formulation allows the model to identify and localize anatomical structures in complex, label-scarce clinical scenarios, and can be instantiated in multiple output formats (VQA, bounding box, and segmentation) while retaining good OOD generalization.

**Strengths:**

**[S1] Clear conceptual novelty in problem formulation (all reviewers).**  The paper proposes a **reference-based learning paradigm** for anatomical understanding: a single labeled reference image is treated as a visually grounded “in-context prior” that guides the VLM to reason about a target image via **spatial relationships** rather than pure text-only medical knowledge or dense supervision.

**[S2] Strong generalization across modalities and structures under limited labels (reviewers 1 and 2).** Experiments show **consistent gains on OOD datasets** (CT → X-ray → ultrasound; organs, bones, vessels), indicating that reference-based spatial reasoning generalizes beyond the training distribution. Initializing segmentation with an RL-tuned VLM further improves OOD performance, providing a concrete recipe for **low-annotation medical pipelines**.

**[S3] Practical relevance in label-scarce clinical scenarios (reviewers 2 and 4).** The framework is explicitly designed for settings where **dense annotations are costly or unavailable**. A small set of high-quality reference templates is sufficient to drive anatomical understanding on large unlabeled cohorts, making the approach appealing for real-world deployment.

**[S4] Solid presentation and implementation clarity (reviewers 2, 3, and 4).** Reviewers 2, 3, and 4 noted that the method is **clearly described**, with a logical layout, carefully designed figures, and detailed reporting of training setup, rewards, and dataset construction, making the work easier to reproduce and extend. And our revised version also addresses the clarity concern of Reviewer 1.

**[S5] Hybrid architecture that cleanly separates reasoning and perception (all reviewers).** By combining a **VLM for relational/spatial reasoning** with **SAM2 for fine-grained segmentation**, RAU achieves precise anatomical region identification while retaining interpretability in terms of reference–target correspondences. The use of special tokens projected into SAM2’s memory further enables a smooth transition from localization to segmentation.



**Response to Concerns:**
The main concerns from the four reviews and our revisions are summarized as follows:

**[C1] Baselines and experimental coverage (medical VLMs and segmentation backbones).**  Reviewers 1, 3 and 4 requested stronger baselines, especially medical VLMs and OOD-robust segmentors. We expanded experiments in two directions:

**1.** VQA / localization: added Lingshu, HuatuoGPT-Vision, MedRegA, BiRD, and MedPLIP.

**2.** Segmentation (VLM+SAM2): added MedSAM2(-Memory/-Ref-SFT), LISA-SFT, and SegZero-SFT. The results (updated Table for segmentation and new VQA tables) consistently show that: (a) richer medical pretraining alone does not solve our anatomical understanding tasks—especially for subtle vessel/bone targets with limited labels; and (b) our reference-based spatial reasoning formulation yields substantially better localization/segmentation, including on OOD datasets (LERA, CAMUS).

**[C2] Retrieval component, reference bank composition, and robustness to reference quality.** Reviewers 1 and 4 asked about reference bank size/composition, retrieval details, and how performance depends on reference quality. In response:

**1.** We emphasized that retrieval was not the conceptual focus of the paper. In real-world deployments, a small number of manually curated high-quality reference templates per site or modality is sufficient. Matching between the test images and reference templates can be achieved by using off-the-shelf or fine-tuned embedding-based retrievers (e.g. DINOv2) or external meta data (e.g. camera parameters in interventional X-ray).

**2.** We added a detailed dataset construction description in the appendix: for each dataset, we build a reference library offline, and each test image is pre-paired with a reference from a different patient with matching label classes; retrieval is only needed at deployment time.

**3.** We added Table A2, which stratifies cases by LPIPS distance between reference and target and reports labeling/segmentation performance. Results show robustness over a wide similarity range, with failures only when the reference is distinctly mismatched (visualized in new failure figures A5/A6).

---

> ### Author Response · Authors · 2025-12-03
>
> **[C3] Clarity, structure, missing implementation details, and citation errors.** Reviewer 1 raised concerns about readability, organization, and rigor. We made substantial structural changes:
>
> **1.** Reorganized Sections 1–3 to separate method vs. experimental setup/results, and to give a clear narrative arc from problem definition to our RAU framework and its three task variants (VQA, bbox, segmentation).
>
> **2.** Added dedicated subsections and an appendix section covering: dataset construction, GRPO reward weights and curriculum, token design/number, memory bank size, and DINOv2+FAISS retrieval configuration.
>
> **3.** Carefully corrected all erroneous citations (e.g., GLIP, SEEM) and manually re-validated references to address concerns about academic rigor.
>
> **[C4] Scope (2D vs 3D) and number of classes.** Reviewers 1 and 4 questioned the use of 3D data and the limited number of classes. We clarified that:
>
> **1.** Our current scope is strictly 2D slice-wise, and extension to 3D/temporal volumes is explicitly positioned as future work.
>
> **2.** We rely on 3D datasets only to obtain sufficiently complex 2D cases; for each test slice, the reference slice is chosen to share the same set of classes.
>
> **3.** The number of classes in the test dataset is non-trivial: RAOS has 20 classes, and Arcade has 26. We acknowledge that in real-world applications, there could be more ROIs; however, due to the lack of such datasets, we are unable to test our method in those conditions.
>
> **[C5] Behavior analysis, failure cases.**  Reviewers 1 and 4 asked to better illustrate how the model reasons and where it fails. We added:
>
> **1.** Figure A3 showing CoT outputs over GRPO training, evidencing a shift toward spatial-relationship reasoning.
>
> **2.** Failure analyses for bbox overlap in slender vessels (Figure A5) and reference-mismatch failures (Figure A6).
>
> **[C6] Novelty and relation to prior work (EVF-SAM, LISA, MedSAM, RAG/ICL).** Reviewers 1 and 4 raised concerns that our architectural contribution was incremental and too close to existing language–segmentation interfaces. In the revision, we **(i)** toned down novelty claims, **(ii)** rewrote the Introduction and Related Work to position our contribution as a task/formulation contribution—reference-based spatial reasoning for anatomical understanding under limited supervision—rather than an architectural one, and **(iii)** added explicit comparisons and discussion with EVF-SAM, LISA, MedSAM2, HuatuoGPT-Vision, Lingshu, MedRegA, BiRD, MedPLIP, and retrieval/ICL-based medical segmentation.
>
> **[Summary]** Overall, the revised manuscript includes the following modifications:
>
>  **(i)** clarification of our main contribution: a new reference-based spatial reasoning formulation for anatomical understanding in complex, label-scarce clinical scenarios;
>
>  **(ii)** extensive new comparisons with strong medical VLMs and segmentation backbones; and
>
>  **(iii)** a more rigorous and transparent exposition of dataset construction, retrieval, and robustness to reference quality.

---

### Meta-Review · Area_Chair_rqpe · 2025-12-30

**Summary:**

Despite the authors making positive changes in their rebuttal, expanding the set of baselines, adding robustness tests, and acknowledging and adjusting the way the contribution is positioned, the core concerns from Reviewer Akfu remain valid, and several major issues are still unresolved. The positioning of the contribution remains unclear, with the core method being highly similar to existing VLM + segmentation fusion / retrieval‑augmented approaches, and the fundamental differences and boundaries from prior work still insufficiently articulated. Significant academic rigor issues are evident, although some citations have been corrected, the serious citation errors in the original manuscript have already undermined its credibility. Baseline fairness and sufficiency are lacking; even with the addition of more models, certain strong methods are not appropriately fine‑tuned for comparison. Scalability and clinical applicability have not been verified. At a top‑tier venue, these constitute significant shortcomings. Multiple reviewers share the view that the work is borderline or below the acceptance threshold.  Given these major concerns, I believe the manuscript is not yet suitable for acceptance and therefore recommend rejection.

**Reviewer Concerns:**

The rebuttal made several progress in expanding baselines, clarifying novelty claims, fixing citation problems, and adding robustness tests.
However, key issues remain unresolved, including the incremental contribution, baseline fairness, underdeveloped retrieval component analysis, and lack of scalability/generalization evidence.

**Reviewer Scores:**

Even with possible slight changes from QGk4 and DJQe, the paper would still average to borderline/reject, because the novelty, evaluation, and academic rigor concerns raised primarily by Akfu remain persuasive to the final evaluation.

---

### Decision · Program_Chairs · 2026-01-26

Reject